# Dataless Weight Disentanglement in Task Arithmetic via Kronecker-Factored Approximate Curvature

**Angelo Porrello**[1]          **Pietro Buzzega**[1]          **Felix Dangel**[2]

**Thomas Sommariva**[1]     **Riccardo Salami**[1]     **Lorenzo Bonicelli**[1]     **Simone Calderara**[1]

[1]University of Modena and Reggio Emilia, Italy
`name.surname@unimore.it`

[2]Vector Institute, Toronto
`fdangel@vectorinstitute.ai`

## Abstract

Task Arithmetic yields a modular, scalable way to adapt foundation models. Combining multiple task vectors, however, can lead to cross-task interference, causing representation drift and degraded performance. Representation drift regularization provides a natural remedy to disentangle task vectors; however, existing approaches typically require external task data, conflicting with modularity and data availability constraints (e.g., privacy requirements). We propose a dataless approach by framing regularization against representation drift as a curvature matrix approximation problem. This allows us to leverage well-established techniques; in particular, we adopt Kronecker-Factored Approximate Curvature and obtain a practical regularizer that achieves state-of-the-art results in task addition and negation. Our method has constant complexity in the number of tasks and promotes robustness to task vector rescaling, eliminating the need for held-out tuning.

## 1 Introduction

Task arithmetic (TA, Ilharco et al., 2022) promises a scalable approach for adapting foundation models. Indeed, fine-tuning produces task-specific parameter updates – called *task vectors* – that can be added or subtracted to edit model behavior. This enables reuse of task-specific knowledge across domains and even backbones (Rinaldi et al., 2025) without retraining. In practice, composing multiple task vectors degrades performance due to cross-task interference: when a new task vector is added, it modifies shared representations, disrupting those used by other tasks. To prevent such interference, task-specific components must be decoupled to preserve other tasks' representations. This property, whereby distinct directions in parameter space lead to changes confined to non-overlapping regions of the input space, is called *weight disentanglement* (Ortiz-Jimenez et al., 2023).

**Encouraging weight disentanglement.** To promote this property, one might regularize the fine-tuning procedure to explicitly preserve other tasks' representations (Yoshida et al., 2025) or, in other words, prevent *representation drift* — i.e., change in a task's activations when new task vectors are added. Nonetheless, such regularizers often require access to other tasks' training data, which is impractical under privacy or regulatory constraints and contradicts modularity and reusability.

> *Therefore, our goal is to design a computationally efficient regularizer for weight disentanglement that can be used without requiring access to the training data.*

This task relates to approximating neural network function space distances (Dhawan et al., 2023), which measure how much a model's behavior changes without requiring access to the original data. Building on this perspective, we incorporate an additional insight specific to TA: fine-tuning the first-order Taylor approximation of the model around its pre-trained parameters empirically enhances weight disentanglement (Ortiz-Jimenez et al., 2023). We show that, under linearization, the representation drift simplifies into a quadratic form of the network Jacobian's *Gramian*, which can be pre-computed on, and shared instead of, the data to enhance weight disentanglement (Fig. 1). However, the Gramian is intractably large, as its size grows quadratically with the number of parameters.

**Link to curvature approximation.** The Jacobian Gram matrix is an instance of the generalized Gauss-Newton (GGN) matrix (Schraudolph, 2003), an extensively studied object in the context of second-order optimization (Martens, 2010; 2020). This link allows us to leverage prior research on

efficient curvature approximations. Specifically, we adopt Kronecker-factored approximate curvature (KFAC, Martens & Grosse, 2015), a block-diagonal approximation of the GGN, where blocks correspond to layers and each block is a *Kronecker product* of two small matrices. KFAC drastically reduces storage and computation while still capturing most intra-layer correlations, bridging the gap between oversimplified diagonal approximations and the intractable full GGN of interest.

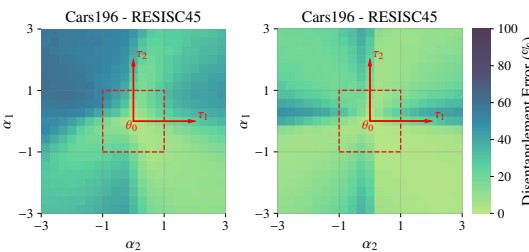

Figure 1: Weight disentanglement *(left)* without and *(right)* with Jacobian Gram regularization.

**Adapting KFAC for TA.** KFAC–based regularization faces a key limitation when applied to multi-task arithmetic: its associated regularizer cannot be accumulated exactly across tasks. The per-task regularizers induce memory and computational costs that grow linearly in the number of tasks. Going beyond the existing approximation, we propose an aggregation scheme that merges per-task curvature factors into a single surrogate, yielding *constant* complexity in the number of tasks.

We show that linking the weight disentanglement objective to curvature-aware optimization yields state-of-the-art performance in *task addition* and *negation* (Ilharco et al., 2022). Furthermore, our method exhibits desirable properties, such as *task localization* – i.e., distinct task vectors govern separate, localized regions in function space associated with different tasks – and *robustness to task vector rescaling*, which renders performance insensitive to scaling coefficients and thus eliminates the need for held-out tuning. In summary, our contributions are the following:

- We derive a regularizer for task arithmetic – called **TAK** (Task Arithmetic with KFAC regularization) – that improves weight disentanglement without using external data.
- We scale representation drift regularization by aggregating per-task regularizers into a single surrogate, ensuring *constant* complexity and storage regardless of the number of tasks.

## 2 BACKGROUND: TASK ARITHMETIC AND LINEARIZED FINE-TUNING

**Setup.** Let $f : \mathbb{R}^D \times \mathbb{R}^P \to \mathbb{R}^C$ denote a neural network that processes a datum $\boldsymbol{x} \in \mathbb{R}^D$ via parameters $\boldsymbol{\theta} \in \mathbb{R}^P$ into a prediction $f(\boldsymbol{x}, \boldsymbol{\theta}) \in \mathbb{R}^C$. During training, these predictions are compared to a target $\boldsymbol{y} \in \mathbb{R}^Y$ via a criterion function $c : \mathbb{R}^C \times \mathbb{R}^Y \to \mathbb{R}$ with the goal to minimize the empirical risk over a training data set $\mathcal{D} = \{(\boldsymbol{x}_n, \boldsymbol{y}_n)\}_n$. We start from a model pre-trained on a large source dataset $\mathcal{D}_0$, yielding pre-trained weights $\boldsymbol{\theta}_0$. Our goal is to fine-tune this model on a specific downstream task $t$ with data set $\mathcal{D}_t$, to obtain the task-specific fine-tuned weights $\boldsymbol{\theta}_t^\star$.

**Task Arithmetic.** The above fine-tuning procedure is typically repeated for multiple ($T$) tasks, yielding *task vectors* $\{\boldsymbol{\tau}_t := \boldsymbol{\theta}_t^\star - \boldsymbol{\theta}_0\}_{t=1}^T$. Such vectors form the core of TA, which posits that simple linear operations in weight space can induce targeted transformations in function space. This enables combining the capabilities of multiple task vectors to build a multi-task model without additional training, through simple linear combination (*task addition*): given the individual task vectors $\{\boldsymbol{\tau}_t\}_{t=1}^T$, the composed model has parameters $\boldsymbol{\theta}_0 + \sum_{t=1}^T \alpha_t \boldsymbol{\tau}_t$ with $\alpha_t \in \mathbb{R}$ (in the simplest case, $\alpha_t = 1$). TA also addresses the removal of task-specific knowledge (*task negation*) by subtracting, rather than adding, a task vector. However, naïve linear composition is prone to interference, as overlapping task-vector updates often conflict and degrade the composed model's performance.

**Linearized fine-tuning.** Ortiz-Jimenez et al. (2023) empirically show that TA benefits from model linearization, particularly when applied during both training and inference. This approach replaces the network with its linear approximation around the pre-trained weights, $(f, \boldsymbol{\theta}_0) \leftrightarrow f_{\text{lin}}$ as

$$f_{\text{lin}}(\boldsymbol{x}, \boldsymbol{\theta}) = f(\boldsymbol{x}, \boldsymbol{\theta}_0) + \mathrm{J}_{\boldsymbol{\theta}} f(\boldsymbol{x}, \boldsymbol{\theta}_0)(\boldsymbol{\theta} - \boldsymbol{\theta}_0), \tag{1}$$

with $\mathrm{J}_{\boldsymbol{\theta}} f(\boldsymbol{x}, \boldsymbol{\theta}_0) \in \mathbb{R}^{C \times P}$ the Jacobian of the model's prediction on datum $\boldsymbol{x}$ with respect to its parameters, evaluated at $\boldsymbol{\theta}_0$. This encourages weight disentanglement in TA, a property whereby task vectors influence the model only on their own tasks, leaving its behavior unchanged elsewhere. Our goal is to construct a regularizer to encourage this property during linearized fine-tuning.

---

**Algorithm 1** Idealized and practical representation drift regularizer for task $t'$

---

**Require:** Network $f(\cdot, \boldsymbol{\theta}_0)$, tasks $\{\mathcal{D}_t\}_{t=1, t \neq t'}^T$
1: Compute per-task GGNs $\{\boldsymbol{G}_{t \neq t'}\}$ (Eq. (3)) (approximate via KFAC, Sec. 3.3)
2: Merge over tasks: $\boldsymbol{G}_{-t'} = \sum_{t \neq t'} \lambda_t \boldsymbol{G}_t$ (optional: merge KFACs, Eq. (8))
3: **return** Quadratic form: $\boldsymbol{\tau} \mapsto \boldsymbol{\tau}^\top \boldsymbol{G}_{-t'} \boldsymbol{\tau}$

---

**Algorithm 2** Linearized FT on task vector $\boldsymbol{\tau}_{t'}$

---

**Require:** Initial weights $\boldsymbol{\theta}_0$, dataset $\mathcal{D}_{t'}$, task vector $\boldsymbol{\tau}_{t'}$ merged curvature matrix $\mathbf{G}_{-t'}$
1: Linearize the net: $(f, \boldsymbol{\theta}_0) \to f_{\text{lin}}(\bullet, \boldsymbol{\tau}_{t'} - \boldsymbol{\theta}_0)$
2: **while** not converged **do**
3:    Draw a mini-batch $\mathcal{B} \sim \mathcal{D}_{t'}$
4:    Minimize objective Eq. (7) on $\mathcal{B}$ w.r.t. $\boldsymbol{\tau}_{t'}$
5: **end while**
6: **return** Task vector $\boldsymbol{\tau}_{t'}$

---

# 3 MAKING REPRESENTATION DRIFT REGULARIZATION DATA-FREE

**Simplified setup with two tasks.** Model linearization simplifies the learning dynamics, allowing us to analyze how editing affects the model. We conduct this analysis in feature space through the lens of *representation drift*, the change in the last-layer activations of a task $t$ when adding a new task $t'$:

$$\left(\substack{\text{Pre-edit} \\ \text{representation}}\right) \ \boldsymbol{z}_t(\boldsymbol{x}) = f_{\text{lin}}(\boldsymbol{x}, \boldsymbol{\theta}_0 + \alpha_t \boldsymbol{\tau}_t) \overset{\text{edit}}{\to} \boldsymbol{z}_{t,t'}(\boldsymbol{x}) = f_{\text{lin}}(\boldsymbol{x}, \boldsymbol{\theta}_0 + \alpha_t \boldsymbol{\tau}_t + \alpha_{t'} \boldsymbol{\tau}_{t'}) \ \left(\substack{\text{Post-edit} \\ \text{representation}}\right)$$

$$\implies \left(\substack{\text{Representation} \\ \text{drift}}\right) \ \Delta_{t \to t, t'}(\boldsymbol{x}) := \|\boldsymbol{z}_{t,t'}(\boldsymbol{x}) - \boldsymbol{z}_t(\boldsymbol{x})\|_2^2 \tag{2}$$

If the drift $\Delta_{t \to t, t'}(\boldsymbol{x})$ vanishes for all $\boldsymbol{x} \in \mathcal{D}_t$, the newly added task vector $\boldsymbol{\tau}_{t'}$ will not interfere as it does not change the model's behavior for inputs from task $t$. Interference between the two tasks can be reduced by penalizing representation drift (Yoshida et al., 2025) via the neural network function space distance (Dhawan et al., 2023) $\mathcal{L}_{t \to t, t'}^{\text{drift}}(\boldsymbol{\tau}_{t'}) := 1/|\mathcal{D}_t| \sum_{\boldsymbol{x} \in \mathcal{D}_t} \Delta_{t \to t, t'}(\boldsymbol{x})$. However, the regularizer for $\boldsymbol{\tau}_{t'}$ requires accessing data of the external task $t$. This may violate segregation policies, impose significant storage demands, and prevent independent training, ultimately reducing flexibility for decentralized training. These issues make direct optimization of this objective impractical in many real-world settings, such as decentralized (McMahan et al., 2017; Kairouz et al., 2021) or privacy-preserving learning scenarios (Abadi et al., 2016; Bonawitz et al., 2017).

## 3.1 CONNECTING REPRESENTATION DRIFT REGULARIZATION TO CURVATURE MATRICES

Now, we reformulate the regularization objective to eliminate its dependence on external task data. Thanks to the linearization, the representation drift from Eq. (2) simplifies into $\Delta_{t \to t, t'}(\boldsymbol{x}) = \|J_{\boldsymbol{\theta}} f_{\text{lin}}(\boldsymbol{x}, \boldsymbol{\theta}_0)(\alpha_t \boldsymbol{\tau}_t - (\alpha_t \boldsymbol{\tau}_t + \alpha_{t'} \boldsymbol{\tau}_{t'}))\|_2^2 = \alpha_{t'}^2 \|J_{\boldsymbol{\theta}} f_{\text{lin}}(\boldsymbol{x}, \boldsymbol{\theta}_0) \boldsymbol{\tau}_{t'}\|_2^2$. The associated regularizer is[1]

$$\boxed{\mathcal{L}_{t \to t, t'}^{\text{drift}}(\boldsymbol{\tau}_{t'}) = \alpha_{t'}^2 \boldsymbol{\tau}_{t'}^\top \boldsymbol{G}_t(\boldsymbol{\theta}_0) \boldsymbol{\tau}_{t'} \ \text{ with } \ \boldsymbol{G}_t(\boldsymbol{\theta}_0) = \frac{1}{|\mathcal{D}_t|} \sum_{\boldsymbol{x} \in \mathcal{D}_t} J_{\boldsymbol{\theta}} f(\boldsymbol{x}, \boldsymbol{\theta}_0)^\top J_{\boldsymbol{\theta}} f(\boldsymbol{x}, \boldsymbol{\theta}_0)} \tag{3}$$

Note that the network Jacobian's Gramian $\boldsymbol{G}_t(\boldsymbol{\theta}_0) \in \mathbb{R}^{P \times P}$ – after initial pre-computation – does not require further data access. This idealized training loop is shown in Alg. 1 (black font).

In exchange for eliminating the data dependency, however, we now face the challenge of computing the $P \times P$ Gramian. This is infeasible even for small neural networks. Thankfully, we can interpret $\boldsymbol{G}_t$ as a curvature matrix that is well-known in the optimization literature: the *generalized Gauss-Newton* (GGN) matrix (Schraudolph, 2003; Martens, 2020). This connection allows us to build on well-established approaches from the optimization literature to efficiently compute structural parametric approximations of $\boldsymbol{G}_t$, ultimately allowing us to make Alg. 1 practical (red font).

## 3.2 THE GENERALIZED GAUSS-NEWTON (GGN) MATRIX

The GGN is a curvature matrix related to the Hessian and arises from partial linearization: The Hessian of a function composition $\ell = c \circ f$ is $\nabla^2 \ell = \nabla^2(c \circ f)$, while the GGN is $\nabla^2(c \circ f_{\text{lin}})$. The standard setting in the second-order optimization literature sets $f$ to be the neural network, and $c$ the criterion function used for training. We now introduce the GGN in this context, showing that the Jacobian Gram matrix from Eq. (3) is an instance of the GGN that results from replacing the training criterion with the squared loss. We can then easily transfer existing GGN approximations.

---

[1]In the following, we suppress $_{\text{lin}}$ since the Jacobians of $f$ and $f_{\text{lin}}$ coincide at $\boldsymbol{\theta}_0$.

**GGN in the training setting.** Consider the neural network $f$ with criterion function $c$ (e.g. cross-entropy) and training data $\mathcal{D}$ from Sec. 2. For sample $n$, define $f_n := f(\bullet, \boldsymbol{x}_n)$ and $c_n := c(\bullet, \boldsymbol{y}_n)$. The example-wise loss is then given by $\ell_n = c_n \circ f_n$, and training minimizes the empirical risk

$$\mathcal{L}(\boldsymbol{\theta}) = \tfrac{1}{|\mathcal{D}|} \sum_n c(f(\boldsymbol{x}_n, \boldsymbol{\theta}), \boldsymbol{y}_n) := \tfrac{1}{|\mathcal{D}|} \sum_n \ell_n(\boldsymbol{\theta}) := \tfrac{1}{|\mathcal{D}|} \sum_n (c_n \circ f_n)(\boldsymbol{\theta}). \tag{4}$$

For brevity, we use $c_n$ to denote the value $c_n(f_n(\boldsymbol{\theta}))$, and $[\bullet]_i$ for slicing (e.g. $[\boldsymbol{a}]_i$ is the $i^{\text{th}}$ entry of $\boldsymbol{a}$). Differentiating the empirical risk twice and applying the chain rule yields the Hessian and its Gauss–Newton decomposition (Schraudolph, 2003; Martens, 2020), containing the GGN $\boldsymbol{G}(\boldsymbol{\theta})$:

$$\nabla^2 \mathcal{L}(\boldsymbol{\theta}) = \boldsymbol{G}(\boldsymbol{\theta}) + \boldsymbol{R}(\boldsymbol{\theta}) := \tfrac{1}{|\mathcal{D}|} \sum_n (\mathrm{J}_{\boldsymbol{\theta}} f_n)^\top \nabla^2 c_n (\mathrm{J}_{\boldsymbol{\theta}} f_n) + \tfrac{1}{|\mathcal{D}|} \sum_n \sum_{m=1}^{C} [\nabla c_n]_m \nabla^2 [f_n]_m. \tag{5}$$

For models that are linear in the parameters, the residual $\boldsymbol{R}(\boldsymbol{\theta})$ vanishes, as it depends on second derivatives, (zero in the linear case). The GGN then coincides with the Hessian of the risk under linearization and, for likelihood-based losses, with the Fisher information matrix (Amari, 2000).

**The Jacobian's Gram matrix as GGN.** The GGN in Eq. (5) generalizes the Jacobian Gram matrix from Eq. (3), used for representation drift regularization, by additionally weighting the Jacobians with the criterion function's Hessian $\nabla^2 c$. If we choose squared error $c_n(\boldsymbol{f}) = 1/2 \|\boldsymbol{f} - \boldsymbol{y}_n\|_2^2$ rather than the training criterion, the GGN becomes the Jacobian Gram matrix exactly, since $\nabla^2 c_n = \boldsymbol{I}_C$.

> *Hence, the matrix $\boldsymbol{G}_t(\boldsymbol{\theta}_0)$ of the quadratic form in Eq.* (3) *corresponds to a curvature matrix: the GGN of the loss $\mathcal{L}(\boldsymbol{\theta})$ (Eq.* (4)*) when the training criterion is the squared loss.*

While the GGN is impractically large to compute or store for neural networks, the literature has developed scalable structured approximations for it. In the following, we build on these approximations (specifically, KFAC) and study how to adapt and extend them in the context of task arithmetic.

### 3.3 KRONECKER-FACTORED APPROXIMATION OF THE GENERALIZED GAUSS-NEWTON

We rely on a structured GGN approximation called *Kronecker-Factored Approximate Curvature* (KFAC) introduced by Martens & Grosse (2015) for fully-connected, then generalized to convolutional (Grosse & Martens, 2016), recurrent (Martens et al., 2018), and transformer architectures (Eschenhagen et al., 2023). KFAC has been successfully applied to optimization (Osawa et al., 2019), pruning (Wang et al., 2019), Laplace approximations (Daxberger et al., 2021; Ritter et al., 2018) and influence functions (Grosse et al., 2023). For an in-depth tutorial, see Dangel et al. (2025).

**Parametric form.** For a net with $L$ layers and parameters $\boldsymbol{\theta}^1, \ldots, \boldsymbol{\theta}^L$, KFAC approximates the GGN as block-diagonal. Each block corresponds to a layer, $\boldsymbol{G}(\boldsymbol{\theta}) = \mathrm{blockdiag}(\boldsymbol{G}(\boldsymbol{\theta}^1), \ldots, \boldsymbol{G}(\boldsymbol{\theta}^L))$, and is further approximated as a Kronecker product, $\boldsymbol{G}(\boldsymbol{\theta}^l) \approx \boldsymbol{B}^l \otimes \boldsymbol{A}^l$. To evaluate the approximation's quadratic form for representation drift regularization, we simply store the Kronecker factors $\{(\boldsymbol{B}_t^l, \boldsymbol{A}_t^l)\}_l$ from task $t$, then evaluate (without instantiating the Kronecker product (Loan, 2000))

$$\mathcal{L}_{t \to t, t'}^{\mathrm{drift}}(\boldsymbol{\tau}_{t'}) = \alpha_{t'}^2 \boldsymbol{\tau}_{t'}^\top \boldsymbol{G}_t(\boldsymbol{\theta}_0) \boldsymbol{\tau}_{t'} \overset{\mathrm{KFAC}}{\approx} \alpha_{t'}^2 \sum_{l=1}^{L} \boldsymbol{\tau}_{t'}^{l\top} (\boldsymbol{B}_t^l \otimes \boldsymbol{A}_t^l) \boldsymbol{\tau}_{t'}^l, \tag{6}$$

with $\boldsymbol{\tau}^l$ denoting the part of $\boldsymbol{\tau}$ corresponding to the parameters in layer $l$.

**KFAC for a single layer.** To illustrate the approximation, consider a single fully-connected layer $l$ in a neural network, with associated weights $\boldsymbol{W}^l \in \mathbb{R}^{D_1 \times D_2}$ (we omit biases for simplicity). The layer processes an intermediate input representation $\boldsymbol{a}_n^l \in \mathbb{R}^{D_2}$ for datum $\boldsymbol{x}_n$ into an intermediate output representation $\boldsymbol{z}_n^l = \boldsymbol{W} \boldsymbol{a}_n^l \in \mathbb{R}^{D_1}$. Further, let $\boldsymbol{\theta}^l := \mathrm{vec}\, \boldsymbol{W}^l \in \mathbb{R}^{D_1 D_2}$ denote the row-flattened weights. The layer's GGN block is $\boldsymbol{G}(\mathrm{vec}\, \boldsymbol{\theta}^l) = 1/|\mathcal{D}| \sum_n (\mathrm{J}_{\boldsymbol{\theta}^l} f_n)^\top \nabla^2 c_n (\mathrm{J}_{\boldsymbol{\theta}^l} f_n)$ and simplifies into a sum of Kronecker products by using the chain rule $\mathrm{J}_{\mathrm{vec}\, \boldsymbol{W}^l} f_n = (\mathrm{J}_{\boldsymbol{z}_n^l} f_n)(\mathrm{J}_{\mathrm{vec}\, \boldsymbol{W}^l} \boldsymbol{z}_n^l)$ where $\mathrm{J}_{\mathrm{vec}\, \boldsymbol{W}^l} \boldsymbol{z}_n^l = \boldsymbol{I}_{D_1} \otimes \boldsymbol{a}_n^{l\top}$ (e.g. Dangel et al., 2020) to obtain

$$\boldsymbol{G}(\mathrm{vec}\, \boldsymbol{W}^l) = \tfrac{1}{|\mathcal{D}|} \sum_n (\mathrm{J}_{\boldsymbol{z}_n^l} f_n)^\top \nabla^2 c_n (\mathrm{J}_{\boldsymbol{z}_n^l} f_n) \otimes \boldsymbol{a}_n^l \boldsymbol{a}_n^{l\top} := \mathbb{E}_n[\boldsymbol{B}_n^l \otimes \boldsymbol{A}_n^l].$$

For the last equality, we use $\mathbb{E}_n[\bullet] = 1/|\mathcal{D}| \sum_n \bullet_n$ for averaging over the data set. KFAC assumes $\mathbb{E}_n[\bullet_n \otimes \star_n] \approx \mathbb{E}_n[\bullet_n] \otimes \mathbb{E}_n[\star_n]$, yielding a single Kronecker product involving the small factors $\boldsymbol{A}^l \in \mathbb{R}^{D_2 \times D_2}$, $\boldsymbol{B}^l \in \mathbb{R}^{D_1 \times D_1}$ to approximate the intractable block $\boldsymbol{G}(\mathrm{vec}\, \boldsymbol{W}^l) \in \mathbb{R}^{D_1 D_2 \times D_1 D_2}$:

$$\boldsymbol{G}(\mathrm{vec}\, \boldsymbol{W}^l) \overset{\mathrm{KFAC}}{\approx} \left( \tfrac{1}{|\mathcal{D}|} \sum_n (\mathrm{J}_{\boldsymbol{z}_n^l} f_n)^\top \nabla^2 c_n (\mathrm{J}_{\boldsymbol{z}_n^l} f_n) \right) \otimes \left( \tfrac{1}{|\mathcal{D}|} \sum_n \boldsymbol{a}_n^l \boldsymbol{a}_n^{l\top} \right) := \boldsymbol{B}^l \otimes \boldsymbol{A}^l.$$

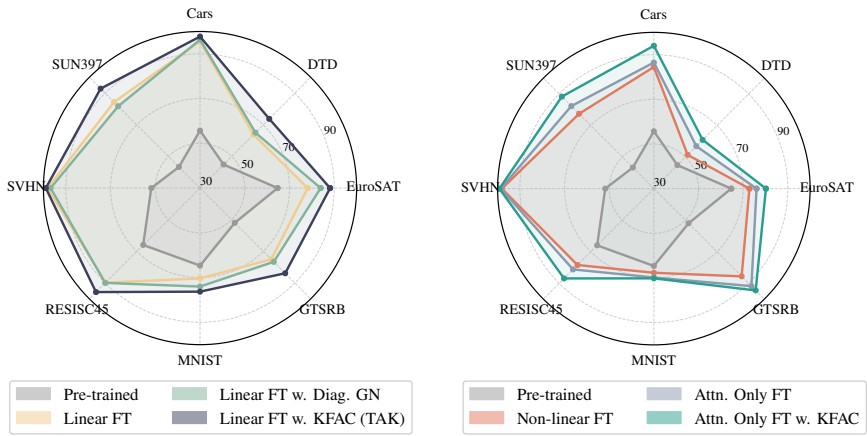

Figure 2: Impact of regularization on "8 Vision" — CLIP ViT-B/16 (abs. accuracy). *Left*: **linearized fine-tuning regime**. *Right*: **non-linear regime**. See the Appendix for CLIP ViT-B/32 and -L/14.

**Variations.** KFAC computes two covariances per layer: (i) the input covariance $\boldsymbol{A}^l = \mathbb{E}_n[\boldsymbol{a}_n^l \boldsymbol{a}_n^{l\top}]$, and (ii) the output gradient covariance $\boldsymbol{B}^l = \mathbb{E}_{n,m}[\boldsymbol{g}_{n,m}^l \boldsymbol{g}_{n,m}^{l\top}]$ of pseudo-gradients $\boldsymbol{g}_{n,m}^l := (\mathrm{J}_{\boldsymbol{z}_n^l} f_n)^\top \boldsymbol{s}_{n,m}$ obtained by backpropagating vectors $\boldsymbol{s}_{n,m} \in \mathbb{R}^C$ related to the Hessian $\nabla^2 c_n$. There exist different variations to compute $\boldsymbol{B}^l$ and – since it is a priori unclear which approach works best in the context of TA – we consider two variants that differ in cost (details in (Dangel et al., 2025)): *(i)* **Exact** (Botev et al., 2017) uses $C$ backpropagations per datum and exactly computes $\boldsymbol{B}^l$; *(ii)* **Monte-Carlo** (MC, Martens & Grosse, 2015) randomizes the exact approach and computes an unbiased MC estimate of $\boldsymbol{B}^l$ using $M < C$ backpropagations per datum (typically, $M = 1$).

### 3.4 MULTI-TASK TRAINING PROCEDURE & REGULARIZATION MERGING

**Naïve multi-task regularization.** While we focused on two tasks, extending to multiple tasks introduces new challenges. To promote disentanglement when training the task vector $\boldsymbol{\tau}_{t'}$, we penalize representation drift with respect to other tasks $t \neq t'$. Starting with the standard training loss $\mathcal{L}_{\mathcal{D}_{t'}}(\boldsymbol{\tau}_{t'}) = 1/|\mathcal{D}_{t'}| \sum_{(\boldsymbol{x}, \boldsymbol{y}) \in \mathcal{D}_{t'}} c(f_{\mathrm{lin}}(\boldsymbol{x}, \boldsymbol{\tau}_{t'} + \boldsymbol{\theta}_0), \boldsymbol{y})$, the overall fine-tuning objective becomes

$$\mathcal{L}_{\mathcal{D}_t}(\boldsymbol{\tau}_{t'}) + \beta \sum_{t \neq t'} \lambda_t \mathcal{L}_{t \to t, t'}^{\mathrm{drift}}(\boldsymbol{\tau}_{t'}) \overset{\mathrm{KFAC}}{\approx} \mathcal{L}_{\mathcal{D}_{t'}}(\boldsymbol{\tau}_{t'}) + \beta \sum_{t \neq t'} \lambda_t \sum_{l=1}^L \boldsymbol{\tau}_{t'}^{l\top} (\boldsymbol{B}_t^l \otimes \boldsymbol{A}_t^l) \boldsymbol{\tau}_{t'}^l , \quad (7)$$

where $\beta$ and $\lambda_t$ control the overall and task-specific regularization strengths, respectively. We weight tasks by data set size, $\lambda_t = |\mathcal{D}_t|/\sum_{t \neq t'} |\mathcal{D}_t|$. Given a pre-computed KFAC of each task $t \neq t'$, this formulation enables regularization without requiring direct access to data sets of external tasks.

**Accumulated regularizer.** A key limitation of the objective in Eq. (7) is that we must store the Kronecker factors individually for each task, incurring $\mathcal{O}(T)$ memory and run time cost. To address this, we build upon the accumulated regularizer $\boldsymbol{G}_{-t'}(\boldsymbol{\theta}_0^l) = \sum_{t \neq t'} \lambda_t \boldsymbol{G}_t(\boldsymbol{\theta}_0^l)$ for layer $l$ and approximate it with a single Kronecker product that captures the contribution of all other tasks:

$$\boldsymbol{G}_{-t'}(\boldsymbol{\theta}_0^l) \overset{\mathrm{KFAC}}{\approx} \sum_{t \neq t'} \lambda_t \boldsymbol{B}_t^l \otimes \boldsymbol{A}_t^l \overset{\mathrm{merge}}{\approx} \left( \sum_{t \neq t'} \boldsymbol{B}_t^l \right) \otimes \left( \sum_{t \neq t'} \lambda_t \boldsymbol{A}_t^l \right). \quad (8)$$

Empirically, this heuristic (Eq. (8)) matches the un-merged formulation's performance (Eq. (7)).

## 4 EXPERIMENTS

**Task addition.** We evaluate performance on the 8 Vision benchmark (Ilharco et al., 2022), which covers eight classification datasets. Using CLIP (Radford et al., 2021) as the foundational vision backbone, we collect eight checkpoints during training for each method and subsequently merge them into a single unified model. Additional details on training and datasets are provided in App. E. Following the original setup (Ortiz-Jimenez et al., 2023), we report both absolute and normalized accuracy. We further analyze the role of the rescaling coefficient $\alpha$: *(i)* setting $\alpha_t = \alpha = 1$ for all tasks, corresponding to plain task-vector addition, and *(ii)* tuning $\alpha$ on a cross-task validation set.

Table 1: **Task addition** results on 8 Vision. The "$\alpha$" column specifies how task vector coefficients are chosen. "1.0" denotes that all coefficients are fixed to 1.0, with no tuning. Numbers marked with † for `TaLoS` (Iurada et al., 2025) are taken from the original paper. See Fig. 2 for a task-wise plot.

| Method | Dataless | $\alpha$ | ViT-B/32 | | ViT-B/16 | | ViT-L/14 | |
| --- | --- | --- | --- | --- | --- | --- | --- | --- |
| | | | Abs. | Norm. | Abs. | Norm. | Abs. | Norm. |
| Pre-trained | – | – | 48.4 | – | 55.4 | – | 65.0 | – |
| Individual | – | – | 90.9 | – | 92.4 | – | 93.8 | – |
| *Linear Fine-Tuning* | | | | | | | | |
| Linear FT | – | 1.0 | 76.7 | 87.2 | 80.2 | 88.9 | 88.0 | 94.8 |
| | – | Best | 78.8 | 89.9 | 82.0 | 90.9 | 88.0 | 94.8 |
| $\tau$Jp (Yoshida et al., 2025) | ✗ | 1.0 | 85.0 | 97.4 | 88.2 | 98.3 | 90.9 | 98.3 |
| | | Best | 85.6 | **98.2** | **88.6** | **98.7** | 91.1 | 98.5 |
| Diag. GGN (Porrello et al., 2025) | ✓ | 1.0 | 80.1 | 92.3 | 82.9 | 93.2 | 87.9 | 96.3 |
| | | Best | 80.2 | 92.5 | 83.0 | 93.3 | 88.0 | 96.4 |
| TAK, **Ours** | ✓ | 1.0 | 85.8 | 97.6 | 88.3 | 97.9 | 91.6 | 99.3 |
| | | Best | **86.0** | 97.8 | 88.3 | 98.1 | **91.6** | 99.3 |
| *Non-Linear Fine-Tuning* | | | | | | | | |
| Non-linear FT | – | 1.0 | 32.0 | 32.9 | 27.4 | 28.2 | 45.3 | 47.5 |
| | – | Best | 73.5 | 80.4 | 77.0 | 82.9 | 84.5 | 89.7 |
| Attn. Only FT (Jin et al., 2025) | – | 1.0 | 22.5 | 23.3 | 22.8 | 23.4 | 66.2 | 69.7 |
| | – | Best | 78.2 | 86.3 | 80.4 | 87.1 | 88.2 | 93.8 |
| `TaLoS`† (Iurada et al., 2025) | ✓ | Best | 79.7 | 90.8 | 82.6 | **92.4** | 88.3 | 95.2 |
| Attn. Only FT + TAK, **Ours** | ✓ | 1.0 | 60.3 | 64.5 | 59.0 | 62.3 | 82.1 | 87.2 |
| | | Best | **83.1** | **91.3** | **84.3** | 91.0 | **89.9** | **95.9** |

**Comparison with related works.** We present a comparative analysis of our regularizer TAK in two distinct regimes. On one hand, we evaluate it in the *linearized regime*, for which it was originally designed; on the other, we examine whether its benefits also extend to the *non-linear regime*. If so, this would broaden the applicability of our approach to most state-of-the-art learning frameworks.

**Linearized fine-tuning regime.** We refer to Fig. 2 (left) for a depiction of the per-task absolute accuracy of the merged model in the linearized regime, while Tab. 1 reports the quantitative results on the 8 Vision benchmark. The results indicate that our KFAC-regularized approach yields substantial improvements against the baseline, achieving performance on par with $\tau$Jp (Yoshida et al., 2025) while avoiding any reliance on external data from other tasks. This makes our method not only more flexible but also inherently privacy-preserving, without sacrificing accuracy. Furthermore, whereas competing methods often require coefficient grid search, TAK proves highly robust: even a simple addition of task vectors ($\alpha = 1$) performs competitively, suggesting that post-hoc tuning can be safely omitted. As a side note, the evidence on ViT-B/32 suggests that the smaller the model scale, the more crucial curvature regularization becomes for achieving strong final performance.

In this setup, we also compare against an approach inspired by Porrello et al. (2025) and apply curvature regularization using a coarse diagonal approximation of the GGN. While both methods exploit curvature information from the pre-trained model, ours relies on KFAC, providing a more accurate estimate that captures intra-layer dependencies. Results show that improved curvature approximations yield larger gains in Task Arithmetic; notably, even diagonal regularization outperforms naïve linear fine-tuning, underscoring the role of regularization in enabling weight disentanglement.

**Non-linear fine-tuning regime.** We now consider the non-linear fine-tuning regime (Tab. 1 and Fig. 2, right). In this setting, alternative approaches attempt to approximate linear behavior without fully linearizing the model. For example, `TaLoS` (Iurada et al., 2025) follows a different route and identifies a subset of parameters that consistently exhibit low gradient sensitivity across tasks and updates only these sparse components. This promotes weight disentanglement during fine-tuning while avoiding the computational bottlenecks of full linearization, enabling efficient task addition and negation. Instead, the authors of Attention-Only Fine-Tuning (Jin et al., 2025) fine-tune only the attention layers of Transformers, showing that this strategy implicitly induces *kernel-like* behavior.

Table 2: **Task negation** on 8 Vision. We report the minimum accuracy on target tasks while preserving at least 95% of the pretrained model's accuracy on control tasks.

| Method | Dataless | ViT-B/32 | | ViT-B/16 | | ViT-L/14 | |
|---|---|---|---|---|---|---|---|
| | | Targ. $\downarrow$ | Cont. $\uparrow$ | Targ. $\downarrow$ | Cont. $\uparrow$ | Targ. $\downarrow$ | Cont. $\uparrow$ |
| Pre-trained | – | 48.4 | 63.3 | 55.4 | 68.3 | 65.0 | 75.5 |
| Non-linear FT | – | 20.4 | 60.5 | 20.4 | 65.3 | 18.1 | 72.4 |
| Linear FT | – | 9.3 | 60.5 | 8.3 | 65.5 | 7.5 | 72.1 |
| TaLoS[†] (Iurada et al., 2025) | ✓ | 11.0 | 60.7 | 10.6 | 66.1 | 10.7 | 73.6 |
| $\tau$Jp (Yoshida et al., 2025) | ✗ | 6.7 | 60.8 | 4.7 | 66.0 | 3.7 | **73.0** |
| TAK, **Ours** | ✓ | **3.4** | **62.4** | **3.4** | **66.4** | **3.5** | 72.6 |

| Method | Dataless | Abs. | Norm. |
|---|---|---|---|
| Individual | – | 85.9 | – |
| MTL | – | 83.6 | – |
| Non-lin. FT | – | 75.7 | 87.7 |
| Linear FT | – | 76.9 | 92.8 |
| Attn. Only FT | – | 72.9 | 85.2 |
| TaLoS | ✓ | 76.3 | 93.4 |
| $\tau$Jp | ✗ | 81.3 | 100 |
| TAK, **Ours** | ✓ | 78.7 | 98.9 |

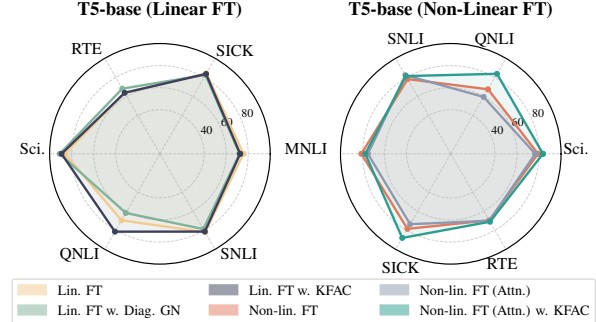

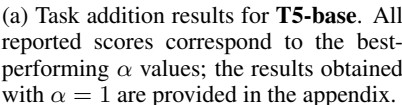

(a) Task addition results for **T5-base**. All reported scores correspond to the best-performing $\alpha$ values; the results obtained with $\alpha = 1$ are provided in the appendix.

(b) Impact of training and regularization choices on language (abs. accuracy). *Left*: linearized regime with no regularization and with the diagonal approximation. *Right* non-linear regime, with attention-only fine-tuning with and without regularization.

Figure 3: Results for language tasks. *Left*: impact of different training strategies and sensitivity to $\alpha$ hyperparameter. *Right*: effects of different regularizations on linear and non-linear fine-tuning.

In this regard, although our regularization is not theoretically exact in the non-linear regime, its applicability can still be justified whenever linearized behavior is implicitly enforced. For this reason, in the non-linear setting we pair our regularizer with Attention-Only Fine-Tuning, which has been shown to induce approximately linear fine-tuning dynamics in Transformers, thereby providing a practical way to extend our method beyond the strictly linearized regime. The results in Fig. 2 (right) show that this is the case: when fine-tuning only attention layers, our approach proves beneficial even in the non-linear regime. Moreover, in this setting, the choice of the $\alpha$ coefficient has a stronger impact on the final accuracy. However, TAK remains the most robust on average, a trend further confirmed by an experiment reported in one of the subsequent paragraphs.

**Unlearning.** We herein investigate a setting where each task vector is subtracted from the pre-trained model. In doing so, we use ImageNet as a control task to verify whether subtraction selectively removes the corresponding task without erasing general knowledge. As shown in Tab. 2, our model achieves stronger forgetting of target tasks while better preserving the control task, surpassing that of the main competitor, $\tau$Jp (Yoshida et al., 2025). Notably, since our regularizer is dataless, it avoids the challenges associated with transferring and storing a "large" data set such as ImageNet to perform regularization. This property is especially promising in the context of the massive data sets used today to train conversational models, where the cost of data access and management is critical.

**Task addition (*language tasks*)** Following Stoica et al. (2025), we test across six natural language tasks: SNLI (Bowman et al., 2015), MultiNLI (Williams et al., 2018), SICK (Marelli et al., 2014), SciTail (Khot et al., 2018), RTE (Wang et al., 2018), and QNLI (Wang et al., 2018), fine-tuning the T5-base model (Raffel et al., 2020). As shown in Fig. 3, TAK consistently outperforms the baselines, particularly under non-linear fine-tuning, thus corroborating the findings observed in vision. However, leveraging data from other tasks ($\tau$Jp) yields additional gains, suggesting that textual domains may still benefit from even more accurate curvature estimation.

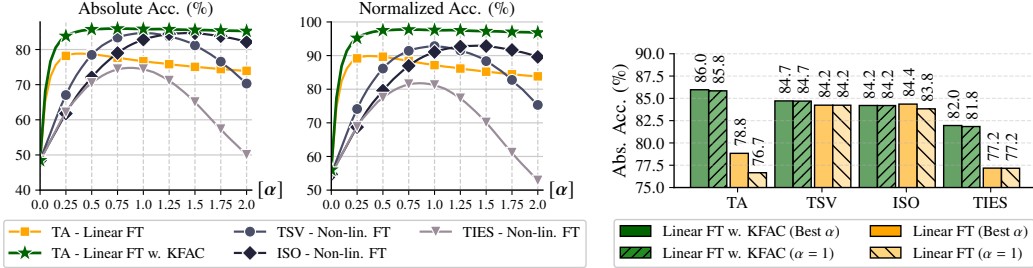

(a) **Model Merging (Non-linear FT)** *vs.* **TA (Linearized FT)**    (b) **Model Merging & Linearized FT**

Figure 4: For ViT-B/32 (8 Vision), we analyze the sensitivity of different merging strategies to the scaling coefficient $\alpha$; a similar analysis for ViT-B/16 is reported in the Appendix. Left: $\alpha$-sweep accuracy of post-hoc merging strategies in the non-linear regime, compared with our linearized and regularized models. Right: performance of merging methods on linearized checkpoints.

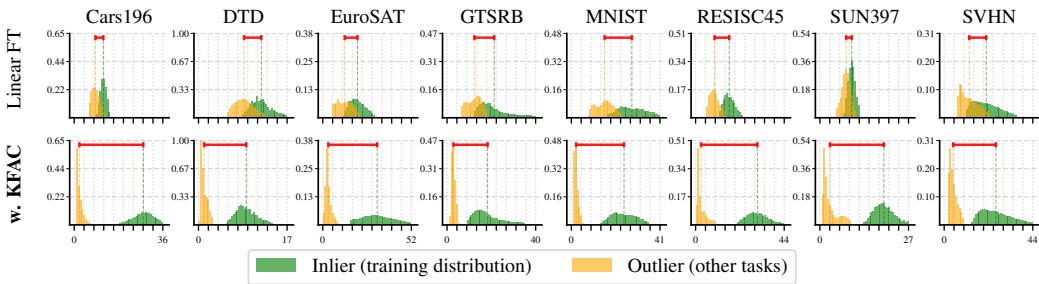

Figure 5: Distribution of $\|\mathrm{J}_{\boldsymbol{\theta}} f(\boldsymbol{x}, \boldsymbol{\theta}_0) \boldsymbol{\tau}_t\|_2^2$ for inputs originating from the training distribution of task $t$ (inliers) versus from other tasks (outliers), under both regularized and non-regularized FT.

**Comparison of model merging strategies.** Fig. 4 compares existing post-hoc approaches for merging task vectors, including TIES (Yadav et al., 2023), TSV (Gargiulo et al., 2025), and ISO (Marczak et al., 2025). We remark that these methods operate after training and are therefore complementary to our approach, which instead acts during training and produces explicitly weight-disentangled task vectors. To assess the benefits of in-training regularization, in Fig. 4a we perform an $\alpha$-sweep over the range $[0, 2]$, focusing on *performance stability* – here, $\alpha$ scales the merged parameters $\boldsymbol{\theta}_0 + \alpha \mathcal{M}(\{\boldsymbol{\tau}_t\}_{t=1}^T)$, where $\mathcal{M}(\cdot)$ denotes the merging strategy. Under KFAC regularization (green curve), simple task-vector summation (Task Arithmetic, TA) achieves the best peak performance and exhibits strong robustness, with accuracy remaining stable over a wide interval of $\alpha$ values. This property makes our approach particularly suitable when $\alpha$ cannot be tuned, e.g., in the absence of a validation set. In practice, this robustness removes the need to access validation data from other tasks, which may be unavailable or undesirable to share. Moreover, as our method TAK relies on simple Task Arithmetic, it avoids expensive operations such as the SVD required by ISO and TSV. As a result, it can be applied in on-the-fly and adaptive model-merging settings (Crisostomi et al., 2026), enabling efficient personalization for specific user requests.

In Fig. 4b, we analyze merging techniques applied to checkpoints obtained in the linearized regime. TA and TIES benefit the most from curvature regularization, whereas ISO and TSV already perform competitively without it. Nevertheless, their performance remains consistently below that of TAK, i.e., Task Arithmetic with curvature regularization. Additional results are reported in App. F.

**Curvature regularization enables Task Localization.** We show that our approach enables a clear separation between training and out-of-distribution examples. Indeed, given an input $\boldsymbol{x}$ and a task vector $\boldsymbol{\tau}_t$, we measure $\|\mathrm{J}_{\boldsymbol{\theta}} f(\boldsymbol{x}, \boldsymbol{\theta}_0) \boldsymbol{\tau}_t\|_2^2$, which we interpret as a *normalcy score* for task $t$. With our regularization (Eq. (3)), these scores are indeed forced to remain low for examples outside the $t$-th training distribution. As illustrated in Fig. 5, this is exactly what we observe in practice: the distribution of $\|\mathrm{J}_{\boldsymbol{\theta}} f(\boldsymbol{x}, \boldsymbol{\theta}_0) \boldsymbol{\tau}_t\|_2^2$ is pushed toward zero whenever the input does not belong to task $t$. With the naïve linear fine-tuning, this behavior is instead much less clear.

Table 3: Our Kronecker-accumulation heuristic *vs.* the idealized multi-task formulation.

| Method | Complexity | $\alpha$ | ViT-B/32 | | ViT-B/16 | | T5-base | |
|---|---|---|---|---|---|---|---|---|
| | | | Abs. | Norm. | Abs. | Norm. | Abs. | Norm. |
| Naïve Multi-Task FT | $\mathcal{O}(T)$ | 1.0 | 86.5 | 98.4 | 88.0 | 97.5 | 78.5 | 97.0 |
| | | Best | 86.6 | 98.5 | 88.1 | 97.6 | 78.5 | 97.0 |
| Accumulated reg. (TAK) | $\mathcal{O}(1)$ | 1.0 | 85.8 | 97.6 | 88.3 | 97.9 | 78.6 | 98.7 |
| | | Best | 86.0 | 97.8 | 88.3 | 98.1 | 78.7 | 98.9 |

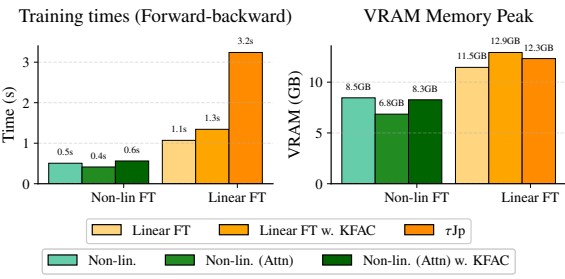

| | Exact | MC=1 (ours) |
|---|---|---|
| $A$ [s] | 1.4 | 1.4 |
| $B$ [s] | 91.5 | 0.2 |
| Total [min] | 198.7 | 3.9 |

(b) Computation time for the KFAC approximation. Reported times for $A$ and $G$ correspond to the *average* over a batch of 8 examples, while the last row shows the total time (in minutes) required to compute the KFAC approximation for all tasks of 8 Vision.

(a) Computational overhead: training times and GPU peak.

Figure 6: Analysis of the overhead of KFAC regularization during training and pre-computation.

This indicates that, under TAK's curvature regularization, each task vector influences the network output only for inputs drawn from its own training distribution. Moreover, this property suggests a natural use of our method for out-of-distribution detection, as it provides a principled mechanism to assess whether an input lies within the model training distribution. A complementary analysis in the non-linear fine-tuning regime is provided in App. F.5, where we compare our method against `TaLoS` and attention-only fine-tuning and observe that the same task-localization behavior persists.

**Naïve multi-task training *vs.* accumulated regularizer.** We herein investigate the impact of the heuristic used in our approach, which accumulates the Kronecker matrices (see Eq. (8)) and thereby avoids a linear cost in the number of tasks. To this end, we run experiments using the idealized naïve multi-task training described in Eq. (7). Our findings, reported in Tab. 3, show that the gap between the idealized and the actual approach is marginal for medium-sized architectures such as ViT-B/16 in vision and T5-base in text. For ViT-B/32, we instead observe a small but consistent gap in favor of the idealized training objective, which aligns with our experience that smaller architectures tend to be more sensitive to curvature regularization and hence to the quality of the approximation.

**Training costs.** Fig. 6 analyzes the overhead introduced by our approach, which is twofold: estimating the KFAC matrices (before training) and computing the regularizer (during training). No overhead is introduced at inference time. With a single Monte Carlo sample, estimating all KFAC matrices for the 8 Vision tasks (128 examples per task) takes **only 4 minutes**, a very limited amount of time compared to the exact approach from Botev et al. (2017). During training, the overhead mainly depends on the chosen regime, with linearized fine-tuning having the largest computational footprint. Nonetheless, KFAC regularization requires only a negligible amount of additional resources, i.e., roughly one third of the training time of $\tau$Jp (Yoshida et al., 2025). This efficiency arises because the $\tau$Jp penalty requires a second forward–backward pass through the (slower) linearized model. Moreover, since TAK does not rely on data for regularization, it avoids the repeated cost of loading new batches into GPU memory, another factor that slows down $\tau$Jp.

**Memory footprint.** Fig. 6 (right) reports the peak VRAM usage across training regimes. KFAC introduces a small increase relative to unregularized baselines: in the linearized regime, it shows a $+12\%$ overhead ($11.5 \rightarrow 12.9$ GB) w.r.t. linear fine-tuning, while in the non-linear attention-only training it shows a $+22\%$ increase ($6.8 \rightarrow 8.3$ GB). For reference, $\tau$Jp peaks at 12.3 GB ($+7\%$ vs. linear FT), and standard non-linear fine-tuning reaches 8.5 GB. No memory overhead incurs at inference since regularization is inactive. Notably, aggregating all per-task KFAC factors into a single surrogate keeps the training footprint of our method at $\mathcal{O}(1)$ w.r.t. the number of tasks.

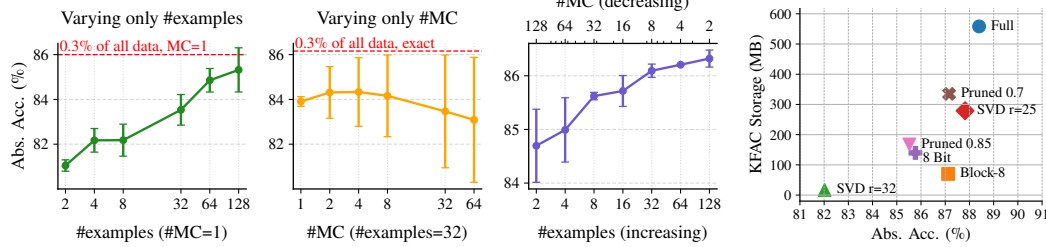

(a) Varying the number of datapoints/Monte Carlo samples for the KFAC. (b) KFAC Storage *vs.* Accuracy.

Figure 7: Effect of KFAC approximation efficiency on performance. Left: impact of the number of datapoints used to estimate the KFAC matrices on downstream accuracy of the merged model. Right: accuracy–memory trade-off induced different KFAC matrix compression strategies.

**KFAC estimation.** In Fig. 7a, we analyze the effect of varying the number of examples and MC samples used for curvature estimation. Our findings (Fig. 7a, Left) indicate that using 128–256 examples is already sufficient to saturate performance, yielding results comparable to those obtained with 30% of each training set. Moreover, final performance is generally on par with that obtained with the exact approximation of Botev et al. (2017). With respect to Monte Carlo sampling, only a few samples per example (1–2) are sufficient. Surprisingly, performance deteriorates beyond this point, with variance across seeds increasing as the number of MC samples grows. Overall, increasing the number of MC samples is less effective than using more data with fewer MC samples.

**KFAC compression.** Unfortunately, the memory cost of storing KFAC matrices scales quadratically with the layer width, which may become challenging for very large models. To mitigate this cost, we evaluate how aggressively KFAC matrices can be compressed – via dynamic 8-bit quantization, structured pruning, block-diagonalization, and truncated SVD (see App. F.6) – without harming accuracy. On ViT-B/16 (8 Vision), these techniques yield substantial memory savings with only minor performance loss (Fig. 7b). The block-based strategy provides the best trade-off, decreasing memory from approximately 550 MB (full KFAC) to about 70 MB – 87% reduction – while incurring only ∼1-point drop in absolute accuracy (88.3 to 87.1).

We additionally analyze whether the KFAC matrices can be moved off-GPU during training without introducing prohibitive overhead. To do so, we evaluate a regime where the penalty loss is computed and backpropagated **only once** every $N$ training steps. As illustrated in Fig. 8, applying the loss every 16 steps leads to a modest degradation (∼1.4 points) relative to applying it at every iteration. This demonstrates that scheduling curvature updates can effectively amortize memory transfers and enable GPU–CPU factor shuffling without compromising the usefulness of the regularizer.

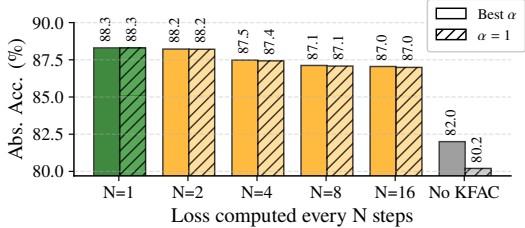

Figure 8: Applying the KFAC loss every N steps.

## 5 CONCLUSIONS

We investigate curvature-based regularization as a means to enhance weight disentanglement in Task Arithmetic and propose TAK (Task Arithmetic with KFAC regularization), a dataless, efficient, and effective approach that makes the simple summation of task vectors competitive with state-of-the-art merging strategies, without additional tuning. We demonstrate applicability in linearized and non-linear regimes, and show that it enables a clear separation between in- and out-of-distribution examples. Our work calls for releasing additional assets together with the pre-trained weights without having to open-source the training data. Such information, e.g. gradient accumulators of the adaptive optimizer used for training (Li et al., 2025), or in our case KFAC, enable further downstream applications with foundation models. Finally, further extending these ideas to models adapted either via standard full or parameter-efficient fine-tuning remains an important direction.

ACKNOWLEDGMENTS

We acknowledge the CINECA award under the ISCRA initiative, for the availability of high-performance computing resources and support. Resources used in preparing this research were provided, in part, by the Province of Ontario, the Government of Canada through CIFAR, and companies sponsoring the Vector Institute. Simone Calderara is supported by the Horizon Europe Chips Joint Undertaking under the NexTArc project (HORIZON-JU-Chips-2024-2-RIA). NexTArc – Next Generation Open Innovations in Trustworthy Embedded AI Architectures for Smart Cities, Mobility and Logistics (Grant Agreement ID: 101194287, DOI: 10.3030/101194287). Additionally, the research activities of Angelo Porrello have been partially supported by the Department of Engineering "Enzo Ferrari" through the program FAR2025DIP (CUP E93C25000370005). We also gratefully acknowledge Symboolic s.r.l. for funding the PhD position of Thomas Sommariva and for the significant contribution of Lorenzo Bonicelli.

REPRODUCIBILITY STATEMENT

To ensure the reproducibility of our results, the complete source code, including model implementations, hyperparameters, and evaluation scripts, is integrated into the Mammoth framework. The codebase will be made publicly available at `https://github.com/aimagelab/mammoth` to support further research and facilitate benchmarking.

DISCLOSURE ON THE USE OF LANGUAGE MODELS

Large Language Models (LLMs) were used exclusively to improve the clarity and polish of the writing. All scientific ideas, methodological contributions, experimental designs, analyses, and conclusions presented in this paper originate entirely from the authors.

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

## A    APPENDIX / SUPPLEMENTARY MATERIAL

The appendix is organized as follows:

- App. B discusses the main limitations of our approach, including memory requirements and curvature-estimation challenges.
- App. C provides a derivation and a formal bound on the approximation error introduced when merging multiple KFAC factors using the Kronecker heuristic.
- App. D presents additional plots illustrating the disentanglement error.
- App. E details the implementation of our methods, with separate discussions for the vision and text domains.
- App. F reports additional experiments. These include:
  - Core analyses:
    * per-task performance analysis,
    * alpha-sweep robustness study (App. F.2),
    * ablation on the regularization coefficient (App. F.3),
    * evaluation of a shared KFAC computed on a reference dataset (App. F.4),
    * task-localization analysis under non-linear fine-tuning (App. F.5);
  - extended experiments:
    * analysis of task localization under memory-efficient KFAC approximations, including block-based, SVD-based, pruning, and 8-bit quantized variants (App. F.6),
    * additional results on more challenging vision domains using a class-incremental partitioning protocol (App. F.7).
- App. G provides a concise overview of prior work on linearized fine-tuning and its recent developments.

## B    LIMITATIONS

KFAC requires storing the Kronecker matrices in GPU memory – two per layer, each with quadratic complexity in the number of units. For large models this can become problematic, suggesting that alternative strategies based on matrix compression or structured Kronecker factors (Grosse et al., 2023; Lin et al., 2024) should be explored. While we combine the well-established KFAC with an accumulation strategy, designing curvature approximations that can easily be merged without sacrificing accuracy may be worth exploring in the future. Moreover, our experiments in the text domain indicate room for improvement, raising the question of whether more sophisticated techniques for curvature estimation could further enhance Task Arithmetic.

## C    APPROXIMATION ERROR OF THE MERGED KFAC FACTORS

For clarity, we focus on a single layer and assume all layers contribute equally, omitting the task weights $\lambda_t$. Let $\{A_t\}_{t=1}^T$ and $\{B_t\}_{t=1}^T$ denote the KFAC factors associated with the tasks involved in the merge. The heuristic used in Eq. 8 replaces the sum of Kronecker products with the Kronecker product between aggregated factors

$$\sum_{t=1}^{T} B_t \otimes A_t \approx \left( \sum_{t=1}^{T} B_t \right) \otimes \left( \frac{1}{T} \sum_{t=1}^{T} A_t \right). \tag{9}$$

We now provide a simple bound that quantifies the error introduced by this approximation. To do so, we define the empirical means and the deviations from the mean

$$\bar{A} = \frac{1}{T} \sum_{t=1}^{T} A_t, \qquad \bar{B} = \frac{1}{T} \sum_{t=1}^{T} B_t, \qquad \Delta A_t = A_t - \bar{A}, \qquad \Delta B_t = B_t - \bar{B}. \tag{10}$$

Note that, by construction, $\sum_t \Delta A_t = \sum_t \Delta B_t = 0$. Substituting $A_t = \bar{A} + \Delta A_t$ and $B_t = \bar{B} + \Delta B_t$ into the left-hand side of Eq. (9) yields

$$\sum_{t=1}^{T} B_t \otimes A_t = \sum_{t=1}^{T} (\bar{B} + \Delta B_t) \otimes (\bar{A} + \Delta A_t) \tag{11}$$

$$= \sum_{t=1}^{T} \left( \bar{B} \otimes \bar{A} + \bar{B} \otimes \Delta A_t + \Delta B_t \otimes \bar{A} + \Delta B_t \otimes \Delta A_t \right) \tag{12}$$

$$= \underbrace{\sum_{t=1}^{T} \bar{B} \otimes \bar{A}}_{T\,\bar{B} \otimes \bar{A}} + \underbrace{\bar{B} \otimes \sum_{t=1}^{T} \Delta A_t}_{=\,0} + \underbrace{\left( \sum_{t=1}^{T} \Delta B_t \right) \otimes \bar{A}}_{=\,0} + \sum_{t=1}^{T} \Delta B_t \otimes \Delta A_t \tag{13}$$

$$= T\,\bar{B} \otimes \bar{A} + \sum_{t=1}^{T} \Delta B_t \otimes \Delta A_t. \tag{14}$$

Substituting $A_t = \bar{A} + \Delta A_t$ and $B_t = \bar{B} + \Delta B_t$ into the right-hand side of Eq. (9), instead, yields

$$\left( \sum_{t=1}^{T} B_t \right) \otimes \left( \sum_{t=1}^{T} A_t \right) = T^2\,\bar{B} \otimes \bar{A}. \tag{15}$$

Hence the approximation error is

$$E := \sum_{t=1}^{T} B_t \otimes A_t \; - \; \frac{1}{T} \left( \sum_{t=1}^{T} B_t \right) \otimes \left( \sum_{t=1}^{T} A_t \right) = \sum_{t=1}^{T} \Delta B_t \otimes \Delta A_t.$$

**Error bound.** Using the Frobenius norm and the property $\|X \otimes Y\|_F = \|X\|_F \|Y\|_F$, we obtain

$$\|E\|_F \leq \sum_{t=1}^{T} \|\Delta B_t\|_F \|\Delta A_t\|_F \leq \sqrt{\sum_{t=1}^{T} \|\Delta B_t\|_F^2} \sqrt{\sum_{t=1}^{T} \|\Delta A_t\|_F^2}. \tag{16}$$

Defining the deviations (standard deviations in matrix space), we obtain:

$$\sigma_A := \sqrt{\frac{1}{T} \sum_{t=1}^{T} \|\Delta A_t\|_F^2}, \qquad \sigma_B := \sqrt{\frac{1}{T} \sum_{t=1}^{T} \|\Delta B_t\|_F^2}, \tag{17}$$

we finally obtain the compact bound

$$\|E\|_F \; \leq \; T\,\sigma_A\,\sigma_B. \tag{18}$$

**Interpretation.** The approximation error is proportional to the product of the variations of the KFAC factors across tasks. When the task-specific factors $(A_t, B_t)$ cluster tightly around their means, both $\sigma_A$ and $\sigma_B$ are small, yielding a small deviation between the true mixed KFAC term and its merged approximation. This situation is particularly likely to occur when the matrices are estimated from a fixed pre-trained backbone such as CLIP: since the underlying feature extractor remains unchanged across tasks, the induced activation and gradient statistics tend to vary only mildly. As a result, the corresponding KFAC factors exhibit limited task-to-task fluctuation, further justifying the accuracy of the merged approximation.

## D   ADDITIONAL PLOTS ON WEIGHT DISENTANGLEMENT

In Fig. 9 we report the disentanglement error, a metric introduced by Ortiz-Jimenez et al. (2023):

$$\xi(\alpha_1, \alpha_2) = \sum_{t=1}^{2} \mathbb{E}_{\boldsymbol{x} \sim \mu_t} \left[ \text{dist} \left( f(\boldsymbol{x}; \boldsymbol{\theta}_0 + \alpha_t \boldsymbol{\tau}_t), f(\boldsymbol{x}; \boldsymbol{\theta}_0 + \alpha_1 \boldsymbol{\tau}_1 + \alpha_2 \boldsymbol{\tau}_2) \right) \right], \tag{19}$$

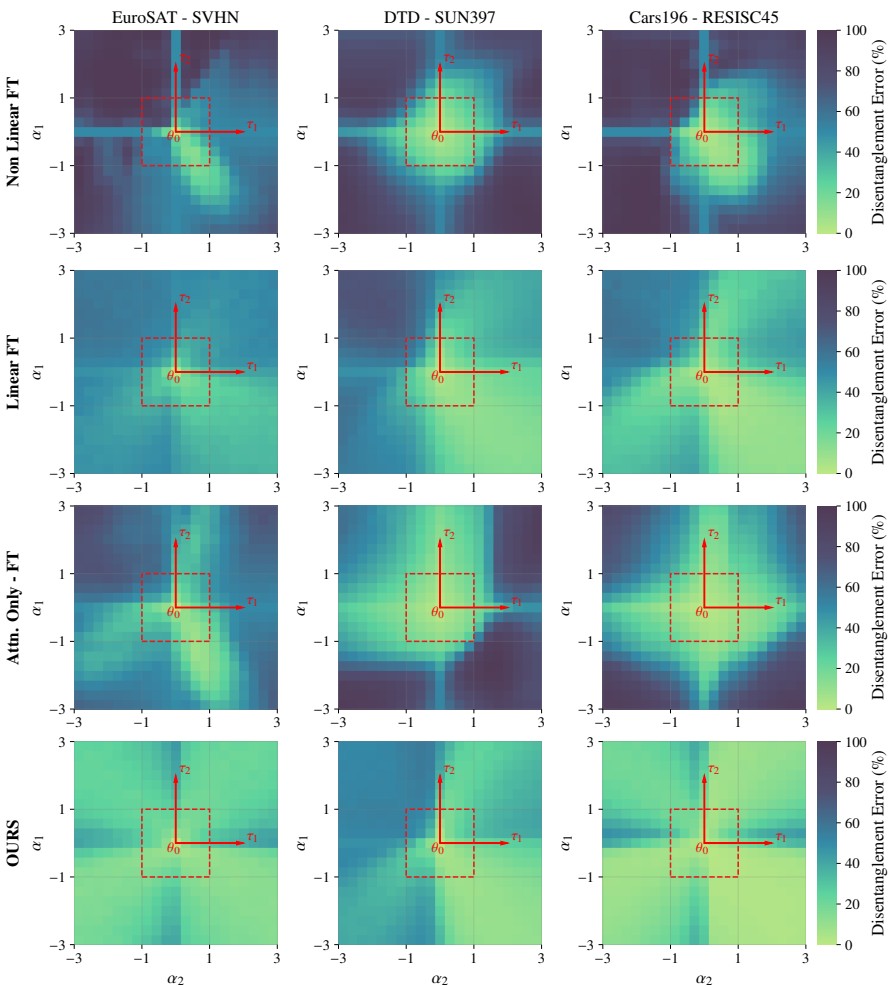

Figure 9: Visualization of weight disentanglement (Ortiz-Jimenez et al., 2023) in ViT-B/16. Non linear fine-tuning Ilharco et al. (2022), Linear fine-tuning Ortiz-Jimenez et al. (2023), Attention-Only fine-tuning Jin et al. (2025), Linear fine-tuning with KFAC regularization.

where $\text{dist}(y_1, y_2) = \mathbb{1}(y_1 \neq y_2)$. When $\xi(\alpha_1, \alpha_2) = 0$, tasks $\tau_1$ and $\tau_2$ merge without interference for the corresponding values of $\alpha_1$ and $\alpha_2$.

As shown in the plots, linearized fine-tuning substantially improves the disentanglement of task vectors. This property is further enhanced under our regularization regime, where only a few darker regions remain, mostly for $\alpha > 1$, a setting that is never used in practice. Notably, in our experiments the disentanglement error is consistently close to zero along the diagonals, which is the most relevant case, since in the literature the common choice is $\alpha_1 = \alpha_2 = \cdots = \alpha_n$.

## E IMPLEMENTATION DETAILS

The GGN information matrices were estimated using a single Monte Carlo sample and computed on 33% of the available training data. However, our empirical analysis showed that sampling only 250-300 training points is sufficient to obtain a reliable estimation of the curvature matrix.

KFAC factors are estimated for all layers involving linear projections in the model – namely, attention and feed-forward projections. In contrast, for LayerNorm parameters and the class token, whose scaling, bias, and token parameters grow linearly rather than quadratically with the embedding dimension, computing the full GGN matrix is tractable. For these components, we therefore use the original, approximation-free GGN instead of its KFAC approximation.

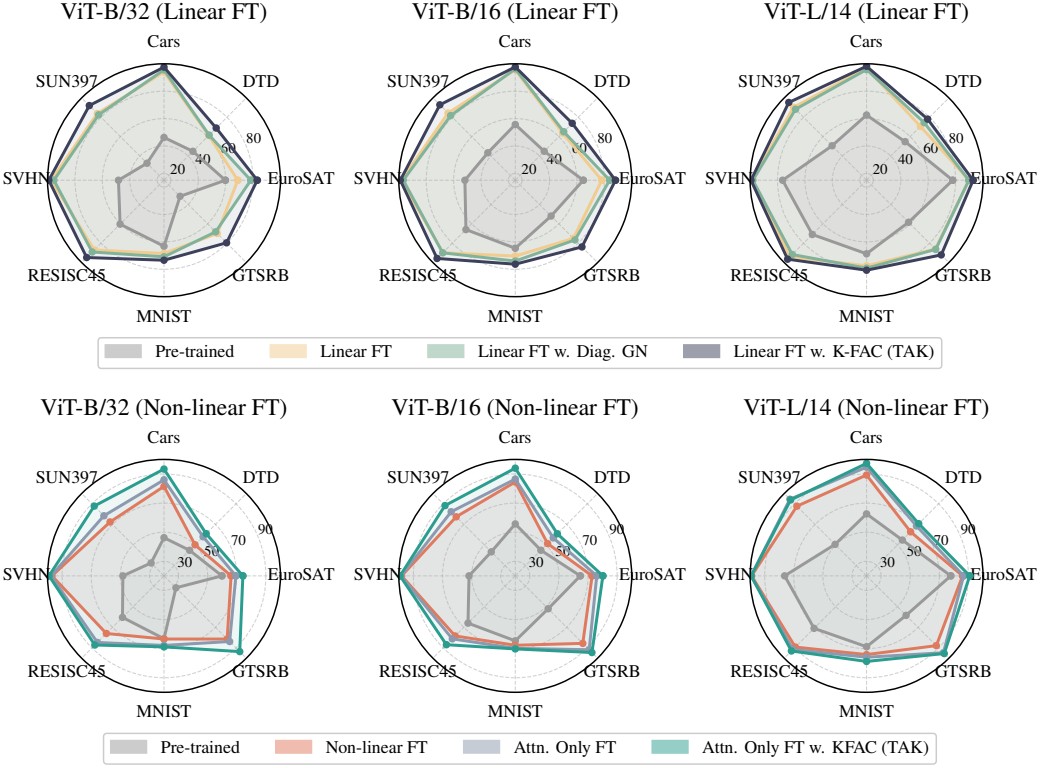

Figure 10: Impact of training and regularization choices on vision tasks (absolute accuracy). Top: linearized regime, compared against the diagonal approximation. Bottom: non-linear regime, compared against attention-only fine-tuning.

The KFAC regularization loss is applied to all fine-tuned layers. Empirically, we found it beneficial to rescale the regularization weight of the last layer of the CLIP visual encoder by a factor of $0.1$.

### E.1 VISION DOMAIN

We leverage the 8 Vision protocol (Ilharco et al., 2022) and conduct experiments on Stanford Cars (Krause et al., 2013), DTD (Cimpoi et al., 2014), EuroSAT (Helber et al., 2019), GTSRB (Stallkamp et al., 2011), MNIST (LeCun et al., 2002), RESISC45 (Cheng et al., 2017), SUN397 (Xiao et al., 2016), and SVHN (Netzer et al., 2011). For training the task vectors, we followed the setup of previous works Ilharco et al. (2022); Ortiz-Jimenez et al. (2023); Yoshida et al. (2025), adopting a batch size of $128$. We used the AdamW optimizer with a learning rate of $3 \times 10^{-4}$, weight decay of $0.1$, and a cosine annealing learning rate scheduler. Unlike prior approaches, we did not apply gradient clipping during training. The regularization term in the loss was weighted by $\lambda = 100$ for ViT-B/32, $\lambda = 500$ for ViT-B/16, and $\lambda = 2000$ for ViT-L/14.

Compared to previous work, we employed a higher learning rate. Since our formulation includes an explicit regularization term in the loss, this allowed us to increase the learning rate without introducing interference across tasks.

### E.2 TEXT DOMAIN

We follow the 6NLI benchmark (Stoica et al., 2025; Panariello et al., 2025), including SNLI (Bowman et al., 2015), MultiNLI (Williams et al., 2018), and SICK (Marelli et al., 2014) which are three-way classification tasks where the relation between a premise and a hypothesis must be identified as entailment, contradiction, or neutral. Additionally, SciTail (Khot et al., 2018), RTE (Wang et al., 2018), and QNLI (Wang et al., 2018) are binary entailment tasks, and therefore fine-tuning

Table 4: Comparison of different merging strategies in the linear fine-tuning regime, with and without KFAC regularization. Results are reported for $\alpha = 1.0$ and the best-performing $\alpha$.

| Method | $\alpha$ | ViT-B/32 | | ViT-B/16 | |
|---|---|---|---|---|---|
| | | Abs. | Norm. | Abs. | Norm. |
| Linear FT + TIES Yadav et al. (2023) | 1.0 | 77.1 | 87.8 | 80.1 | 88.7 |
| | Best | 77.2 | 87.8 | 80.3 | 88.9 |
| Linear FT + TSV Gargiulo et al. (2025) | 1.0 | 84.2 | 96.3 | 86.7 | 96.5 |
| | Best | 84.2 | 96.3 | 86.8 | 96.6 |
| Linear FT + ISO Marczak et al. (2025) | 1.0 | 83.8 | 95.8 | 86.9 | 96.7 |
| | Best | 84.4 | 96.4 | 87.3 | 97.1 |
| TAK, **Ours** + TIES Yadav et al. (2023) | 1.0 | 81.8 | 92.7 | 86.6 | 95.9 |
| | Best | 82.0 | 92.8 | 87.1 | 96.6 |
| TAK, **Ours** + TSV Gargiulo et al. (2025) | 1.0 | 84.7 | 96.3 | 87.6 | 97.2 |
| | Best | 84.7 | 96.3 | 87.7 | 97.3 |
| TAK, **Ours** + ISO Marczak et al. (2025) | 1.0 | 84.2 | 95.6 | 87.2 | 96.8 |
| | Best | 84.2 | 95.6 | 87.3 | 96.9 |

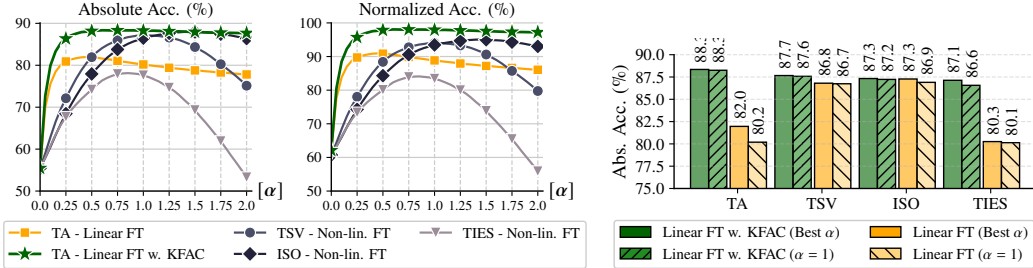

(a) **Model Merging (Non-linear FT)** *vs.* **TA (Linearized FT)**    (b) **Model Merging & Linearized FT**

Figure 11: For ViT-B/16 (8 Vision), we analyze the sensitivity of different merging strategies to the scaling coefficient $\alpha$. Left: $\alpha$-sweep accuracy of post-hoc merging strategies in the non-linear regime, compared with our linearized and regularized models. Right: performance of merging methods on linearized checkpoints.

and evaluation are restricted to two labels. For training language task vectors, we adopted a batch size of 128, using an AdamW optimizer with a learning rate of $3 \times 10^{-4}$ with an iteration-based cosine-annealing scheduler and a weight decay of 0.01. Like in vision tasks, we did not apply gradient clipping during training. The regularization term in the loss is set to $\lambda = 20$ for the KFAC regularization and to $\lambda = 0.1$ for the diagonal regularization.

## F  ADDITIONAL EXPERIMENTS

In this section we present the results of additional experiments on task addition conducted on the 8 Vision benchmark, complementing those already reported in the main paper.

### F.1  PERFORMANCE

Fig. 10 provides a per-task breakdown of the same experiment reported in Tab. 1. Interestingly, the larger ViT-L/14 backbone exhibits smaller relative gains from regularization, particularly in the non-linear regime, where its behavior closely resembles that of its linearized counterpart. Consistent with prior work Ortiz-Jimenez et al. (2023), this suggests that very large models may already display an implicit form of regularization. Conversely, the ViT-B/32 benefits the most from regularization, showing that smaller architectures require more careful fine-tuning to enable effective task arithmetic.

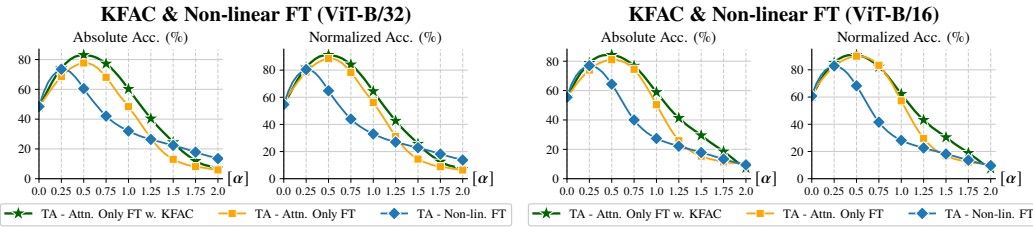

(a) ViT-B/32, $\alpha$-sweep comparison.

(b) ViT-B/16, $\alpha$-sweep comparison.

Figure 12: Sensitivity to the scaling coefficient $\alpha$ in the non-linear fine-tuning regime. We report $\alpha$-sweep results for ViT-B/32 (left) and ViT-B/16 (right), comparing standard non-linear fine-tuning, attention-only fine-tuning Jin et al. (2025), and its variant regularized with the KFAC.

Table 5: On 8 Vision, ablation of $\lambda$ on ViT-B/32 (left) and ViT-B/16 (right). All performances are reported in terms of absolute accuracy using $\alpha = 1$.

| | ViT-B/32 | | | | | ViT-B/16 | | | |
|---|---|---|---|---|---|---|---|---|---|
| $\lambda$ | **Seed** 7 | **Seed** 21 | **Seed** 42 | **AVG.** | $\lambda$ | **Seed** 7 | **Seed** 21 | **Seed** 42 | **AVG.** |
| 0 | 75.0 | 75.4 | 75.1 | $75.2 \pm 0.028$ | 0 | 79.1 | 78.7 | 79.1 | $79.0 \pm 0.188$ |
| 1 | 82.2 | 82.4 | 80.6 | $81.7 \pm 0.648$ | 1 | 83.2 | 83.4 | 83.8 | $83.5 \pm 0.265$ |
| 10 | 85.2 | 85.1 | 85.1 | $85.1 \pm 0.002$ | 50 | 86.9 | 86.8 | 87.0 | $86.9 \pm 0.059$ |
| 100 | 86.2 | 85.8 | 86.0 | $86.0 \pm 0.026$ | 500 | 88.0 | 87.9 | 88.2 | $88.0 \pm 0.114$ |
| 1000 | 86.5 | 86.4 | 86.4 | $86.4 \pm 0.002$ | 5000 | 88.3 | 88.4 | 88.4 | $88.4 \pm 0.015$ |
| 10000 | 84.5 | 84.4 | 84.3 | $84.4 \pm 0.006$ | 50000 | 86.7 | 86.6 | 86.6 | $86.6 \pm 0.002$ |

## F.2 ROBUSTNESS UNDER TASK ARITHMETIC: ALPHA-SWEEP ANALYSIS

In Fig. 11, we extend the analysis presented in the main paper to the ViT-B/16 backbone. The same trends observed for ViT-B/32 hold also in this setting, confirming the consistency of our findings across model scales. For completeness, we additionally report in Tab. 4 the explicit performance of the different model merging strategies evaluated in the linearized regime.

We then conduct a similar $\alpha$-sweep analysis focusing on the application of our method in the non-linear fine-tuning regime. As shown in Fig. 12, across both ViT-B/32 and ViT-B/16, attention-only fine-tuning Jin et al. (2025) and its KFAC-regularized variant exhibit increased robustness to variations of the scaling coefficient $\alpha$ compared to standard non-linear fine-tuning, with our method achieving both higher peak performance and improved robustness. However, when compared to the analyses in Figs. 4 and 11, which examine the linearized and KFAC-regularized model (i.e., TAK), the non-linear regime remains significantly more sensitive to $\alpha$, suggesting an intrinsic advantage of approaches that combine linearization with disentanglement-aware regularization.

## F.3 ABLATION ON THE REGULARIZATION COEFFICIENT

This section presents an ablation study investigating the impact of the scaling coefficient $\lambda$ applied to the regularization term in the loss function. In Tab. 5 we evaluate the performance of ViT-B/32 and ViT-B/16 using six values of the regularization coefficient, ranging over five orders of magnitude from 0 to $10^4$, and repeated each experiment with three random seeds. The case $\lambda = 0$ serves as the baseline, corresponding to non-regularized fine-tuning. It should be noted that these results differ from those reported in Tab. 1, as the linear fine-tuning therein follows the hyperparameter configuration of Ilharco et al. (2022), whereas the experiments presented here employ the hyperparameter setting described in App. E.

The results indicate that the proposed method is robust with respect to the choice of $\lambda$. Optimal performance is observed for values of $\lambda$ between $10^2$ and $10^3$, while only minor degradation occurs for $\lambda = 10$ and $\lambda = 10^4$. This behavior confirms that successful model merging primarily depends on the presence of regularization based on information from the generalized Gauss-Newton matrix,

Table 6: Task addition results on the eight vision datasets when using either task-specific KFAC factors or a single shared KFAC computed on ImageNet-1k. Results show that a universal, task-agnostic KFAC (ImageNet-KFAC) retains most of the benefits of our regularizer while requiring no access to auxiliary task-specific data.

| Method | Dataless | $\alpha$ | ViT-B/32 | | ViT-B/16 | |
|---|---|---|---|---|---|---|
| | | | Abs. | Norm. | Abs. | Norm. |
| Linear FT | − | 1.0 | 76.7 | 87.2 | 80.2 | 88.9 |
| | − | Best | 78.8 | 89.9 | 82.0 | 90.9 |
| TAK, **Ours** | ✓ | 1.0 | **85.8** | **97.6** | **88.3** | **97.9** |
| | | Best | **86.0** | **97.8** | **88.3** | **98.1** |
| ImageNet-TAK, **Ours** | ✓ | 1.0 | 84.7 | 97.0 | 86.0 | 95.4 |
| | | Best | 84.7 | 97.0 | 86.0 | 95.4 |

and that the magnitude of this term must be sufficiently emphasized. However, the results also show that no precise tuning of $\lambda$ is required to achieve strong performance.

## F.4 ELIMINATING TASK DEPENDENCE WITH A UNIVERSAL KFAC

Although our framework completely removes the need for raw auxiliary data, it still requires pre-computed input and gradient covariance factors from the tasks to be disentangled. This dependence may be limiting in scenarios where such factors cannot be shared due to practical difficulties in storing or distributing task-specific curvature statistics, or simply because the set of tasks to be composed is not known in advance at training time.

To assess whether this dependence can be relaxed, we test whether broad curvature statistics – extracted from a large, natural-image distribution – can serve as a proxy and effectively replace the per-task KFAC factors. In details, we build a variant, denoted *ImageNet-KFAC*, in which every layer uses a single pair of $A/B$ matrices computed on ImageNet-1k. Ideally, these factors capture universal visual covariances, and hence they can remain fixed for all downstream tasks. During fine-tuning, these shared factors can entirely substitute the task-specific ones normally employed by our regularizer.

As shown in Tab. 6, despite using non–task-specific information, this proxy KFAC recovers approximately 97–99% of the performance obtained with full task-specific factors on both ViT-B/16 and ViT-B/32 (8 Vision). The absolute accuracy reached by the ImageNet-KFAC variant is 84.7% on ViT-B/32 and 86.0% on ViT-B/16, closely matching the performance of the original approach while substantially surpassing diagonal or no-regularization baselines as well as competitive alternatives such as `TaLoS` or attention-only fine-tuning.

These results indicate that a task-agnostic curvature prior, captured by a single shared factorization, delivers most of the benefits of our dataless regularizer without accessing any task-specific statistics. In practical scenarios, this makes the method fully data-agnostic with respect to the problem, effectively eliminating any residual coupling to external tasks.

## F.5 TASK LOCALIZATION UNDER NON-LINEAR FINE-TUNING

In this section we extend the task-localization analysis presented in the main paper to the non-linear fine-tuning regime. The goal is to assess whether the separation between in-task and out-of-task examples, induced by our curvature regularizer under linearized training, persists when full model parameters are updated. In details, we measure the same editing-localization metric used in the main paper, namely the difference between the Jacobian-projected output variation $\|\mathrm{J}_{\boldsymbol{\theta}} f(\boldsymbol{x}, \boldsymbol{\theta}_0)\, \boldsymbol{\tau}_t\|_2^2$ for inputs belonging to task $t$ versus those coming from other tasks.

As shown in Fig. 13, we evaluate four methods: the standard non-linear fine-tuning, `TaLoS` Iurada et al. (2025), attention-only fine-tuning Jin et al. (2025), and our proposed KFAC-based curvature regularizer. For each approach, we fine-tune the model in the fully non-linear setting and compute the distribution of normalcy scores for in-task and out-of-task inputs.

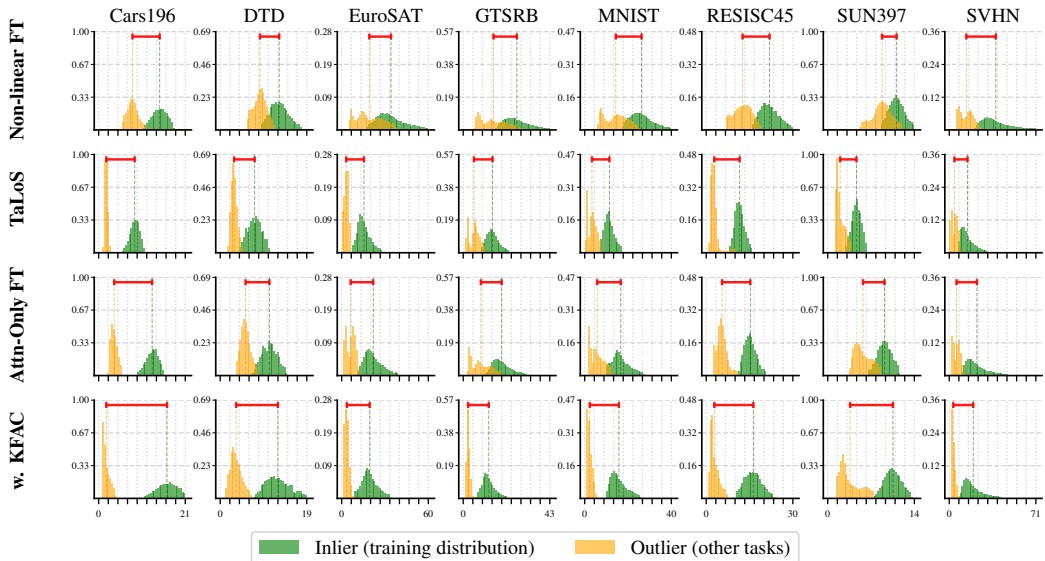

Figure 13: Task localization under **non-linear fine-tuning**. We report the distribution of the Jacobian-projected normalcy scores $\|\mathrm{J}_{\boldsymbol{\theta}} f(\boldsymbol{x}, \boldsymbol{\theta}_0)\,\boldsymbol{\tau}_t\|_2^2$ for inputs belonging to task $t$ (in-task) versus inputs from all other tasks (out-of-task).

The results show a consistent pattern across all datasets. Our method maintains a clear and sharp separation between in-distribution and out-of-distribution examples, closely mirroring the behavior observed under the linearized regime. `TaLoS` and attention-only fine-tuning preserve part of this effect but yields a weaker distinction. Overall, these findings confirm that curvature regularization continues to restrict the influence of each task vector to its corresponding training distribution even when the full network is fine-tuned.

### F.6  KFAC COMPRESSION STRATEGIES AND TASK LOCALIZATION

To assess the robustness of our curvature regularizer under memory constraints, we evaluate several compression strategies applied directly to the KFAC factors. All strategies described below are applied independently to both $A$ and $B$ matrices for every layer.

The first strategy is a **block-diagonal approximation** ("Block 8"), in which each factor is partitioned into eight equally sized blocks along the main diagonal, with all off-diagonal blocks discarded. This yields a substantial reduction in memory while maintaining a structured representation and preserving dominant second-order interactions.

The second strategy relies on **truncated SVD**. Given the factorization $A = U\Sigma V^\top$, we keep only the top singular components, either by selecting a fixed rank (32 in our experiments) or by retaining a percentage of the original rank (25%). The truncated reconstruction $\tilde{A} = U_k\Sigma_k V_k^\top$ provides a low-rank surrogate that preserves the principal curvature directions.

A third strategy applies unstructured **magnitude pruning**. Each KFAC matrix is converted to COO sparse format, and only the largest-magnitude entries are preserved. We consider two keep ratios, 30% and 15%, corresponding to increasingly aggressive sparsification. All remaining entries are set to zero, effectively reducing memory and bandwidth requirements.

Finally, we evaluate **dynamic 8-bit quantization**. Each factor is quantized on-the-fly to an 8-bit integer representation, with per-row scaling ensuring that reconstruction errors remain controlled.

**Task localization.** We further investigate whether the task-localization behavior observed in the main paper remains stable when applying memory-efficient KFAC approximations. In particular, we focus on the block-based compression strategy, where each KFAC factor is decomposed into 8 diagonal blocks, substantially reducing storage while preserving the structure of the Kronecker

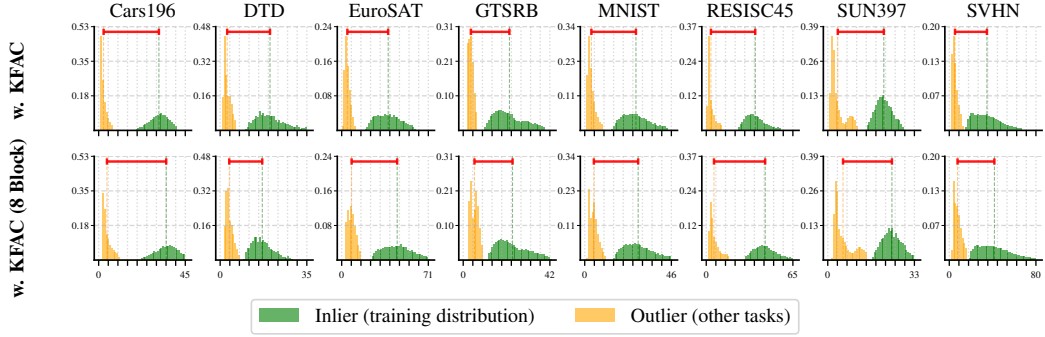

Figure 14: Task localization under linearized fine-tuning with block-compressed KFAC. The separation between the two distributions closely matches that of the full KFAC model, indicating that the block-based compression has negligible impact on task localization and that curvature-based task isolation remains robust even under aggressive memory reductions.

Table 7: Performance comparison across different regularization strategies on ViT-B/16

| Model | ImageNet-R | EUROSAT | RESISC45 |
|---|---|---|---|
| Zero-shot | 77.72 | 49.48 | 66.02 |
| Non-linear FT | 82.32 | 71.21 | 73.85 |
| Linear FT | 81.66 | 70.40 | 72.28 |
| Linear FT w. Diag. GN | 81.64 | 73.94 | 74.04 |
| $\tau$jP Yoshida et al. (2025) | 81.28 | 84.36 | 84.83 |
| TAK, **Ours** (naive penalty) | 82.64 | 79.64 | 78.91 |
| TAK, **Ours** (aggregated penalty) | 82.63 | 79.64 | 78.30 |

approximation. This variant is the most promising among those we evaluated, as it consistently provides the best trade-off between memory savings and accuracy.

The results, shown in Fig. 14, reveal that the block-based KFAC approximation preserves the same localization behavior as the full KFAC model. Even with only eight diagonal blocks per factor, the model continues to sharply distinguish in-distribution from out-of-distribution samples. The compression therefore appears to have negligible impact on this diagnostic, suggesting that curvature-based task localization is robust to coarse, memory-friendly KFAC approximations.

### F.7 EXPERIMENT ON OTHER VISION DOMAINS

In Tab. 7 we present additional experiments on a different vision domain to further assess the effectiveness of KFAC regularization on less trivial tasks. Following (Porrello et al., 2025), each dataset is split into partitions containing distinct classes. This procedure ensures task diversity while keeping the domain consistent, since all partitions originate from the same dataset. The number of classes per partition depends on the dataset: ImageNet-R (Hendrycks et al., 2021) is divided into 10 tasks of 20 classes each, RESISC45 (Krizhevsky & Hinton, 2009) into 9 tasks of 5 classes each, and EuroSAT (Helber et al., 2019) into 5 tasks of 2 classes each. After fine-tuning the base model on each partition, the resulting models are merged and evaluated on the full test set, considering the union of all classes across tasks rather than restricting evaluation to the classes of the training task only, as done in the 8 Vision benchmark. Accuracy is then reported on this joint classification problem, following the protocol of (Porrello et al., 2025). These experiments demonstrate that KFAC regularization achieves state-of-the-art performance even under this more challenging setting.

| Method | Dataless | Abs. | Norm. |
|--------|----------|------|-------|
| Individual | – | 85.9 | – |
| Non-lin. FT | – | 65.5 | 75.9 |
| Linear FT | – | 76.1 | 92.0 |
| Attn-Only FT | – | 67.0 | 78.3 |
| TaLoS | ✓ | 75.8 | 92.8 |
| $\tau$Jp | ✗ | **81.0** | **99.5** |
| KFAC, **Ours** | ✓ | 78.6 | 98.7 |

Table 8: Task addition results for **T5-base** with $\alpha = 1$.

### F.8 TEXT DOMAIN: RESULTS FOR $\alpha = 1$

**Results for $\alpha = 1$.** We follow the setup described in the main text for language tasks and evaluate T5-base using the fixed hyperparameter value $\alpha = 1$. As reported in Tab. 8, our method exhibits consistently strong performance in the text domain, mirroring the trends observed in the vision setting.

## G RELATED WORKS ON LINEARIZED FINE-TUNING

Linearized models offer a principled lens for analyzing fine-tuning by considering first-order expansions around a pre-trained initialization. Foundational work (Arora et al., 2019; Jacot et al., 2018) showed that infinitely wide networks trained with gradient descent follow kernel gradient flow under the Neural Tangent Kernel (NTK), yielding exact functional characterizations of training dynamics. This perspective has since been extended to more realistic settings, including representation learning (Mu et al., 2020), small-data regimes (Arora et al., 2020), and random-matrix studies of generalization (Wei et al., 2022). Building on these insights, several linearized fine-tuning approaches have been proposed to improve efficiency and stability, such as LQF (Achille et al., 2021), privacy-preserving updates (Golatkar et al., 2021), improved task-head initialization (Ren et al., 2023), continual learning (Shon et al., 2022), and language-model adaptation (Malladi et al., 2023). More recent work explores model composition and ensembling through tangent-space operations (Liu & Soatto, 2023; Tang et al., 2024).

The linearized regime has also become central to task arithmetic. Tangent-space representations have been linked to weight disentanglement and reliable task editing (Ortiz-Jimenez et al., 2023; Porrello et al., 2025; Yoshida et al., 2025; Liu et al., 2024). Within this framework, NTK-based approximations enhance task separability and make linear combinations of task vectors more predictable, further underscoring the versatility of model linearization for fine-tuning, composition, and editing.

