# OpenReview forum: "Dataless Weight Disentanglement in Task Arithmetic via Kronecker-Factored Approximate Curvature"
_ICLR.cc/2026/Conference — ICLR 2026 Poster_

### Official Review · Reviewer_K5Dr · 2025-10-17

**Soundness:** 3
**Presentation:** 3
**Contribution:** 3
**Rating:** 6
**Confidence:** 4

**Summary:**

This paper proposes a new method for weight disentanglement in task arithmetic. To address the limitations of prior constraints on cross-task representation drift—which often require access to auxiliary data during fine-tuning and incur high computational cost—the authors approximate the penalty via Kronecker-factored decomposition. This lets fine-tuning proceed without explicit data, needing only each layer’s input covariance and output-gradient covariance, while substantially reducing compute. Experiments across both vision and NLP tasks show that the approach matches or surpasses the strong baseline $\tau$Jp.

**Strengths:**

- Clear presentation: The paper is well structured, easy to follow, and provides accessible explanations of the theoretical derivations.

- Novelty of bringing KFAC into task arithmetic: It reframes the traditional representation-drift penalty as a curvature (GGN) approximation under squared loss, then applies a KFAC approximation to make the method more data-efficient during fine-tuning and significantly cheaper computationally.

- Comprehensive empirical evaluation: Beyond merge performance versus recent state-of-the-art methods, the experiments also cover cost analysis, comparisons with linearization-based approximations, and ablations over different KFAC estimation choices.

**Weaknesses:**

- Dependence on other-task statistics (A, B): While the method mitigates the need for raw auxiliary data required by τJp, it still relies on access to precomputed input/gradient covariance factors from the other tasks to be disentangled. This dependence may limit applicability in settings with privacy or access constraints and introduces coupling to external task-specific artifacts.

- Missing non-linear regime comparisons to SOTA: The paper compares against recent methods such as TSV and ISO only in the linear regime, not in the non-linear regime where those methods are primarily positioned. If performance is comparable to—or worse than—those SOTA approaches, the practical advantage of the proposed method remains unclear, especially since TSV/ISO can merge using parameters alone whereas the proposed approach requires auxiliary task statistics.

- Underspecified regularization strength selection: The KFAC regularizer’s weight is fixed per task/dataset, but the paper does not clearly describe how these values were chosen nor provide sensitivity or robustness analyses. This gap raises concerns about reproducibility and about how performance depends on the strength of the regularization.

**Questions:**

See weaknesses

---

> ### Author Response · Authors · 2025-11-21
>
> We thank Reviewer K5Dr for the careful and insightful review, and for the encouraging remarks on the contribution of bringing KFAC into Task Arithmetic. The points raised have helped us clarify the practical scope and limitations of our approach, and we have expanded both the experiments and the discussion accordingly.
>
> ## Dependence on other-task statistics (A, B)
>
> > While the method mitigates the need for raw auxiliary data required by $\tau$Jp, it still relies on access to precomputed input/gradient covariance factors from the other tasks to be disentangled. This dependence may limit applicability in settings with privacy or access constraints and introduces coupling to external task-specific artifacts.
>
> We understand the reviewer’s concern and fully acknowledge the criticism. However, **we can show that this limitation can be mitigated in practical scenarios**. Indeed, the task-specific KFAC factors (A and B matrices) can be replaced with factors computed once on a generic support dataset containing *universal* visual statistics. For example, **estimating KFAC factors on ImageNet-1k** can effectively simulate the curvature induced by broad, real-world data distributions. To clarify this point, we added subsection F.4 in the updated Appendix, where we evaluate a ‘universal KFAC’ variant—using a single pair of ImageNet-1k A/B factors per layer.
>
> | Method | ViT/B-32 Abs. Acc. | ViT/B-32 Norm. Acc. | ViT/B-16 Abs. Acc. | ViT/B-16 Norm. Acc. |
> |--------|---------|----------|---------|----------|
> | A-B ImageNet-1k ($\alpha$=1) | 84.7% | 97.0% | 86.0% | 95.4% |
> | A-B ImageNet-1k ($\alpha$ best) | 84.7% | 97.0% | 86.0% | 95.4% |
> | A-B from other tasks ($\alpha$=1) | 86.0% | 97.7% | 88.4% | 98.0% |
> | A-B from other tasks ($\alpha$ best) | 86.1% | 97.8% | 88.4% | 98.0% |
>
> On ViT-B/16 and ViT-B/32, this proxy recovers $\approx$ 97–99% of the original performance and still provides a clear advantage over existing regularization strategies. For comparison, the ImageNet-KFAC variant reaches 84.7% (ViT-B/32) and 86.0% (ViT-B/16) absolute accuracy, whereas TaLoS achieves only 79.67 (ViT-B/32) and 82.60 (ViT-B/16), and attention-only fine-tuning yields 78.2 (ViT-B/32) and 80.4 (ViT-B/16).
>
> This demonstrates that **even a task-agnostic curvature prior retains most of the benefits of our dataless regularizer** while maintaining a significant performance margin over strong baselines. We thank the reviewer for highlighting such an important issue and for giving us the opportunity to improve the manuscript.
>
> ## Missing non-linear regime comparisons to SOTA
>
> > The paper compares against recent methods such as TSV and ISO only in the linear regime, not in the non-linear regime where those methods are primarily positioned. If performance is comparable to $-$ or worse than $-$ those SOTA approaches, the practical advantage of the proposed method remains unclear, especially since TSV/ISO can merge using parameters alone whereas the proposed approach requires auxiliary task statistics.
>
> We would like to clarify why our approach remains practically advantageous even when compared with strong post-hoc methods. In brief, our curvature-regularized training produces task vectors that are explicitly **weight-disentangled** and substantially **more robust** to the choice of $\alpha$ (i.e., the scaling coefficient applied to the merged task vector). As we show a few lines below, the resulting performance is on par with state-of-the-art post-hoc approaches, while offering a set of capabilities that post-hoc methods cannot support.
>
> **Weight disentanglement.** Our work focuses on *Task Arithmetic*, not model merging. This distinction is crucial: task arithmetic requires more than combining models; it requires meaningful operations on task vectors. Our method enables both addition *and* subtraction, i.e., zero-shot unlearning of pretraining concepts by subtracting task vectors. Post-hoc nonlinear merging methods such as TSV and TIES do not produce disentangled task vectors and cannot support effective task negation, as they operate on entangled nonlinear updates.
>
> **Robustness to the choice of $\alpha$.** As shown in the paper, our approach yields task vectors that are far more robust to the choice of $\alpha$ $-$ the scaling coefficient used during merging (see introductory paragraph of Sec. 4). This robustness removes the need for validation data from *each* task, whereas post-hoc merging methods such as TSV explicitly require per-task validation to tune $\alpha$ effectively. In practical settings where external validation data are unavailable, this makes our approach considerably more reliable.

---

> > ### Author Response · Authors · 2025-11-21
> >
> > Nevertheless, following the reviewer’s feedback, **we carried out an explicit comparison between our method and a state-of-the-art post-hoc approach such as TSV**. In particular, we evaluated both methods over a wide range of $\alpha$ values (from $0$ to $1.5$) to assess not only their peak performance but also their robustness to the choice of the scaling coefficient.
> >
> > ### Alpha Sweep TSV ViT-B/16 absolute accuracy
> > | METHOD \ $\alpha$ | 0.1 | 0.2 | 0.3 | 0.4 | 0.5 | 0.6 | 0.7 | 0.8 | 0.9 | 1.0 | 1.1 | 1.2 | 1.3 | 1.4 | 1.5 |
> > | --- | --- | --- | --- | --- | --- | --- | --- | --- | --- | --- | --- | --- | --- | --- | --- |
> > | TSV - Original implementation | 61.4| 68.4|74.0|78.4 |81.8|84.5|86.5|87.8| 88.4|88.8|88.6|88.4|87.8|87.1|86.0|
> > | TSV - Our implementation | 62.7|69.5 | 74.5| 78.8| 81.9| 84.0|85.4 |86.3 | 86.8|87.1|87.0| 86.7|86.3| 85.5| 84.3|
> > | Ours - KFAC |79.7|84.9 | 86.8|87.7| 88.0|88.3 | 88.4|88.4 |88.4 |88.4| 88.4| 88.4|88.4 | 88.4| 88.4|
> >
> > ### Alpha Sweep TSV ViT-B/16 normalized accuracy
> > | METHOD \ $\alpha$ | 0.1 | 0.2 | 0.3 | 0.4 | 0.5 | 0.6 | 0.7 | 0.8 | 0.9 | 1.0 | 1.1 | 1.2 | 1.3 | 1.4 | 1.5 |
> > | --- | --- | --- | --- | --- | --- | --- | --- | --- | --- | --- | --- | --- | --- | --- | --- |
> > | TSV - Original implementation |67.1 | 74.4|80.2| 84.9|88.6|91.5|93.3|95.1| 95.7|96.2|96.0|95.7|95.0|94.1|92.7|
> > | TSV - Our implementation | 68.3| 75.3| 80.5|85.1 |88.4|90.7 |92.2|93.2|93.7|94.0|93.9|93.5|92.9|92.0|90.7|
> > | Ours - KFAC | 88.0|93.8|96.0 | 97.1|97.5|97.8 |97.9|97.9| 98.0|98.0|97.9 | 97.9| 97.9| 97.9| 97.9|
> >
> > For transparency, we report both the results reported in the original TSV paper and those obtained with our own re-implementation. It is noted that the two TSV variants yielded slightly different results, and we believe this discrepancy is due to environment incompatibilities (our framework cannot run on the older PyTorch version used in their implementation).
> >
> > Across the full $\alpha$-sweep, our KFAC regularization consistently outperforms our re-implementation of TSV. When compared to the original TSV results, the picture is similar: while peak accuracy is similar once $\alpha$ is tuned, our method stays strong over a much broader range of $\alpha$ values. This robustness matters in practice, because **it removes the need for a cross-task validation set** to run a grid-search $-$ a requirement that post-hoc merging methods like TSV still depend on.
> >
> > By contrast, our approach is inherently dataless in this regard as well: it never requires transferring samples across compute nodes. This makes it substantially easier to deploy in distributed settings.
> >
> > ## Underspecified regularization strength selection
> >
> > We now provide a sensitivity analysis w.r.t. the choice of $\lambda$, showing that the method is stable across a wide range of values (see Sec. F.3 in the updated Appendix). In practice, performance varies minimally within one order of magnitude around the chosen coefficient, indicating that the method is not sensitive to precise hyperparameter tuning. In our experiments, any value in the range $\lambda \in [10, 1000]$ consistently yields strong performance.
> >
> > ### **ViT-B/32**
> > |$\lambda$|seed 7|seed 21|seed 42|Avg. $\pm$ std.|
> > |-|-|-|-|-|
> > |0|75.0|75.4|75.1|75.2 $\pm$ 0.028|
> > |1|82.2|82.4|80.6|81.7 $\pm$ 0.648|
> > |10|85.2|85.1|85.1|85.1 $\pm$ 0.002|
> > |100|86.2|85.8|86.0|86.0 $\pm$ 0.026|
> > |1000|86.5|86.4|86.4|86.4 $\pm$ 0.002|
> > |10000|84.5|84.4|84.3|84.4 $\pm$ 0.006|
> >
> > ### **ViT-B/16**
> > |$\lambda$|seed 7|seed 21|seed 42|Avg. $\pm$ std.|
> > |-|-|-|-|-|
> > |0|79.1|78.7|79.1|79.0 $\pm$ 0.188|
> > |1|83.2|83.4|83.8|83.5 $\pm$ 0.265|
> > |50|86.9|86.8|87.0|86.9 $\pm$ 0.059|
> > |500|88.0|87.9|88.2|88.0 $\pm$ 0.114|
> > |5000|88.3|88.4|88.4|88.4 $\pm$ 0.015|
> > |50000|86.7|86.6|86.6|86.6 $\pm$ 0.002|

---

> > > ### Comment · Reviewer_K5Dr · 2025-11-25
> > > **Reply to authors**
> > >
> > > Thank you for addressing my concern thoughtfully.
> > > I still have questions regarding the universal KFAC.
> > >
> > > You adopt ImageNet as a “universal” source, and while I agree it covers general imagery, it likely does not support medical images or highly niche domains. In such cases, I believe limitations of the proposed approach would remain. Do you think this scenario is fundamentally difficult, or is regularization feasible using alternative data drawn from a different distribution? If the latter, how would such alternative data be obtained, and could you include practical experimental validation to support this?

---

> > > > ### Author Response · Authors · 2025-11-25
> > > >
> > > > Thank you for raising this point; it is indeed an important scenario to consider. However, we believe that the proposed regularization **remains fully feasible** even under such settings. Below, we provide empirical evidence supporting this claim.
> > > >
> > > > In situations where K-FAC factors are unavailable and the ImageNet assumption no longer holds, we argue that the most practical strategy is to compute the K-FAC matrices **using the closest publicly available dataset** in terms of both appearance and high-level visual cues. To illustrate this, we conduct an experiment on the **ISIC 2018 Challenge** [b] benchmark, which focuses on the classification of skin lesions from **dermoscopic images**. Following the experimental setup introduced in [a], we split the dataset into three tasks; afterwards, the goal is to compute a task vector on each task independently.
> > > >
> > > > Now, suppose that we cannot rely on the K-FAC matrices extracted from the other tasks. In this case, a practical solution is to search for the most visually similar publicly available dataset and compute the K-FAC matrices on this external source. We identified **Derm7pt** [c], a small but curated dataset of 1,011 images for skin-lesion diagnosis. We emphasize that ISIC and Derm7pt are completely disjoint datasets, published in different articles by different groups, and we verified that they do not share images.
> > > >
> > > > In this experiment, we hence compare four approaches: the base model without regularization (baseline), the ImageNet-KFAC variant, the practical Derm7pt-KFAC variant, and our model regularized with K-FAC from other ISIC tasks (upper bound). We evaluate all methods on the ViT-B/16 architectures.
> > > >
> > > > | Method                               | Acc. Abs ($\alpha$ best) | Acc. Abs  ($\alpha=1$) |
> > > > |--------------------------------------|--------------|----------------------|
> > > > | Non-Linear FT (no reg.) |90.3|89.5|
> > > > | Linear FT (no reg.) |90.9|90.9|
> > > > | Linear FT + A-B ImageNet-1k |94.2|93.0|
> > > > | Linear FT + A-B Derm7pt |95.5|95.2|
> > > > | Linear FT + A-B from other ISIC tasks (upper bound)   |**96.9**|**96.9**|
> > > >
> > > > As can be seen, although K-FAC computed on the other ISIC tasks remains the best-performing option (last row), the practical strategy of estimating K-FAC on a related dataset such as Derm7pt yields **very competitive performance**, and is substantially better than using no regularization at all. As rightly pointed out by the reviewer, the ImageNet-KFAC variant is less effective; nevertheless, it still delivers a **clear improvement over the baseline**, indicating that it remains a viable and reliable option even in such specialized domains.
> > > >
> > > > It is worth noting that a similar experiment could be conducted in the satellite-imagery domain using EuroSAT and RESISC as target and support datasets, respectively (or viceversa). If deemed useful, we can provide these additional results as well, and we would be glad to ask the reviewer whether they consider it important to include these experiments (both medical and satellite) in the main paper or in the appendix.
> > > >
> > > > We remain fully available to address any further questions or clarifications the reviewer may have.
> > > >
> > > > [a] Menabue, M., et al. (2024, September). Semantic residual prompts for continual learning. In European Conference on Computer Vision (pp. 1-18). Cham: Springer Nature Switzerland.
> > > >
> > > > [b] Codella, N. C., et al. (2018, April). Skin lesion analysis toward melanoma detection: A challenge at the 2017 international symposium on biomedical imaging (isbi), hosted by the international skin imaging collaboration (isic). In 2018 IEEE 15th international symposium on biomedical imaging (ISBI 2018) (pp. 168-172). IEEE.
> > > >
> > > > [c] Kawahara, J., et al. (2018). Seven-point checklist and skin lesion classification using multitask multimodal neural nets. IEEE journal of biomedical and health informatics, 23(2), 538-546.

---

> > > > > ### Comment · Reviewer_K5Dr · 2025-11-26
> > > > > **Reply to Authors**
> > > > >
> > > > > Thank you for the thorough response to my additional concerns.
> > > > >
> > > > > All of my concerns are now resolved, and I feel more positive about acceptance, so I have increased my score.
> > > > >
> > > > > Thank you for the excellent work and the careful rebuttal.

---

> > > > > > ### Author Response · Authors · 2025-11-26
> > > > > >
> > > > > > We are glad to hear that our clarifications fully addressed your concerns, and we sincerely thank you for your thoughtful follow-up.

---

### Official Review · Reviewer_RdRs · 2025-10-31

**Soundness:** 4
**Presentation:** 4
**Contribution:** 3
**Rating:** 6
**Confidence:** 4

**Summary:**

This paper introduces a data-free approach to improving task arithmetic grounded in Kronecker-Factored Approximate Curvature (K-FAC), a second-order optimization approximation that captures local curvature information in parameter space.

**Strengths:**

1. Quality: Good. The paper presents a well-motivated and technically sound contribution, with strong theoretical grounding in second-order optimization and empirical evidence supporting its claims. Experimental evaluation is relatively comprehensive, covering diverse architectures and tasks, with consistent metrics and clear ablation studies isolating the contribution of curvature-based disentanglement.
2. Clarity: The paper is very well written and easy to read.
3. Significance: The proposed solution is efficient and effective (on par with the best provided TA method for vision datasets, yet slightly lower performance on language datasets). Its performance on both standard finetuning and linear finetuning shows its flexibility.

**Weaknesses:**

1. Minor Novelty concern because this paper seems relatively incremental as it’s built upon the key ideas of Ortiz-Jimenez et al., 2023 and Yoshida et al. (2025).

**Questions:**

1. I got confused by the naming of “Dataless” because even linear finetuning requires data, and gradients/Jacobian require data.
2. In Eq(5), is $R(\theta)$ also removed in nonlinear finetuning?
3. Can you discuss the relationship of GGN and other Fisher or Hessian matrix based methods like “Efficient Model Editing with Task-Localized Sparse Fine-tuning.” (Iurada, Leonardo, et al. 2025) and “Merging Models with Fisher-Weighted Averaging” (Matena, Raffel, 2022)?
4. Notation: In line 179, what does the coefficient matrix refer to here? In line 218, what is $s_{n,m}$?
5. Eq 8: What’s the memory and runtime cost of this approximation and can you explain any intuition of this approximation?
6. In Figure 5, wonder why some of the 8 standard vision tasks like SVHN are missing.
7. In Table 1 & 2, include the results of Porrello et al. (2025)’s numbers for a more straightforward comparison.
8. Line 369 “naive” typo.

---

> ### Author Response · Authors · 2025-11-21
>
> We thank Reviewer RdRs for the positive and detailed assessment of our work, in particular regarding its clarity, technical soundness, and empirical evaluation.
>
> ## Weakness: Novelty
>
> First, we would like to clarify the meaning of the term dataless, as its interpretation may have caused confusion regarding the novelty of our contribution.
>
> > I got confused by the naming of "Dataless". Like any fine-tuning approach, our method uses data from the current task during training.
>
> The term "dataless" specifically refers to **our proposed regularization component**, not to the fine-tuning process itself. Other methods (*e.g.*, τJp) require sharing other tasks' data to measure representation drift, whereas our formulation leverages precomputed **KFAC matrices** that can be exchanged without exposing any samples or labels.
>
> **Novelty.** Our work builds upon prior foundations but introduces several key innovations. First, as noted by reviewer K5Dr, we bring KFAC into Task Arithmetic by **reframing the representation-drift penalty as a curvature (GGN) approximation** under squared loss. On top of this contribution, we apply a KFAC approximation to make the method substantially "dataless", more data-efficient and computationally lightweight. Building on this perspective, our contribution adds a *constant-complexity multi-task scaling* scheme, achieved by aggregating per-task KFAC factors into a single surrogate, ensuring that the regularizer remains independent of the number of tasks.
>
> ## Question: Eq(5) and R(θ)
>
> > In Eq(5), is R(θ) also removed in nonlinear finetuning?
>
> In the nonlinear fine-tuning setting (which we also evaluate), the term $R(\theta)$ in Eq. (5) is **not** zero and does **not** vanish in principle. However, in all our nonlinear experiments, following the observations of Jin et al. (2025), we explicitly enforce a *linearized regime* during fine-tuning by updating only the attention layers. Under this restricted update regime, the model behaves similarly to its linearized counterpart, and the contribution of $R(\theta)$ becomes negligible. In other words, although $R(\theta)$ is theoretically non-zero for nonlinear fine-tuning, such constrained optimization keeps the updates similar to those in the linear regime, making our approximation valid in practice.
>
> ## Question: Relation with Other Works
>
> > Can you discuss the relationship of GGN and other Fisher or Hessian matrix based methods like “Efficient Model Editing with Task-Localized Sparse Fine-tuning.” (Iurada, Leonardo, et al. 2025) and “Merging Models with Fisher-Weighted Averaging” (Matena, Raffel, 2022)?
>
> The works cited (Iurada et al., 2025; Matena & Raffel, 2022) are indeed related, as they also rely on curvature information. However, both compute the diagonal Fisher Information Matrix $-$ *i.e.*, an approximation to the Hessian of the classification loss used for training. Their curvature therefore reflects the properties of the task-specific supervised objective.
>
> Our setting differs in two key ways. First, although the Gauss–Newton matrix (GGN) can always be tied to the Hessian of some loss, in our case this loss is *not* the classification objective: it corresponds instead to an MSE regression over the penultimate-layer features of the pre-trained backbone. As a result, our curvature captures the geometry of the **representation space before the classifier**, rather than the curvature of the downstream task. This aligns with the original interpretation of **task arithmetic as a property of the predictors** (Ortiz-Jimenez et al., 2023), not of the fine-tuned objectives.
>
> Second, unlike Matena & Raffel, our curvature is always estimated on the **fixed pre-trained model**, whereas theirs is computed **after training**, at a local optimum of each task-specific fine-tuning (here, the Fisher–Hessian equivalence holds). Moreover, we employ a **K-FAC structured approximation**, while they use a **diagonal** one.
>
> A similar distinction applies to TaLoS. Their method also extracts curvature-like information from the pre-trained network, but through the diagonal cross-entropy Fisher (gradient sensitivity). Since the parameters of the model are not at (or near) a minimum of the classification loss, the usual connection between the Fisher information matrix and the true Hessian no longer holds. As a result, the curvature estimate becomes substantially noisier, which explains why their method requires multiple refinement rounds to stabilize the approximation. As before, we use a structured **KFAC** approximation rather than a diagonal one, yielding a richer and more stable curvature signal.

---

> > ### Author Response · Authors · 2025-11-21
> >
> > ## Question: Notation
> >
> > > In line 179, what does the coefficient matrix refer to here?
> >
> > The coefficient matrix in line 179 refers to the task-specific curvature matrix $G_t(\theta_0)$. In the revised version of the manuscript, we have updated the text to make this point explicit and avoid any possible ambiguity.
> >
> > > In line 218, what is $s_{n,m}$?
> >
> > $s_{n,m}$ denotes the vector associated with data point $n$ and Monte-Carlo sample $m$ that is backpropagated through the network to estimate the GGN. We noticed that this was not stated explicitly in the main text and will add the clarification.
> >
> > ## Question: Memory and Runtime Cost
> >
> > > Eq. (8): what’s the memory and runtime cost of this approximation?
> >
> > In summary, the approximation does not introduce any additional runtime overhead, while memory usage is substantially reduced, since only a single set of K-FAC factors needs to be stored.
> >
> > First, the approximation does not alter how KFAC is used during training: as in standard KFAC, we never materialize the full Kronecker product. The only difference is that, instead of storing one pair of factors per task, we aggregate them into a single pair. As a result, the memory footprint becomes identical to storing a single KFAC matrix, regardless of the number of tasks. Without the approximation, we would need to keep all task-specific factors in memory, which scales linearly with the number of tasks; with Eq. (8), memory stays constant.
> >
> > Second, the runtime impact is negligible. Aggregating the factors only requires summing the existing matrices once, before training. This cost is trivial compared to computing one K-FAC penalty per task, each of which requires a full forward–backward pass at every training step. With our aproximation, once the aggregated factors are formed, all subsequent operations have exactly the same cost as regularizing against a single pair of K-FAC matrices.
> >
> > ## Question: Missing Figures
> >
> > > In Figure 5, wonder why some of the 8 standard vision tasks like SVHN are missing.
> >
> > We omitted SUN-397 and SVHN from Fig. 5 of the original submission **solely for aesthetic reasons**. In the updated version of the manuscript, we now include all eight datasets without excluding any of them, ensuring full transparency (see Fig. 5).
> >
> > ## Question: Table editing
> >
> > > Table 1 \& 2: include the results of (Porrello 2025).
> >
> > These results were previously available only in the supplementary materials. In the updated version of the manuscript, we have now included them directly in the main tables for completeness and clarity.
> >
> > ## Question: Line 369 “naive” typo.
> >
> > Thank you for catching this. We corrected the typo in the newer version of our paper.

---

### Official Review · Reviewer_1ihd · 2025-11-01

**Soundness:** 3
**Presentation:** 3
**Contribution:** 3
**Rating:** 4
**Confidence:** 4

**Summary:**

The paper proposes a data-free regularizer for Task Arithmetic (TA) that penalizes cross-task representation drift using a curvature surrogate. Starting from the linearized network around the pre-trained weights, the drift penalty becomes a quadratic form in the Jacobian Gram / GGN matrix; the authors approximate it with KFAC, yielding a per-layer Kronecker structure that is cheap to evaluate during fine-tuning (Eqs. (3), (6)). They further introduce a merged regularizer that aggregates per-task KFAC factors into a single surrogate to keep cost constant in the number of tasks (Eq. (8)). Experiments on CLIP (8-Vision) and T5 show strong gains for TA addition/negation and competitiveness with τ-JP—without needing other tasks’ data. Notably, ViT-L/14 reaches 91.6 abs / 99.3 norm with α=1, and their method excels at task negation while preserving control accuracy. The paper also claims minimal training overhead and code release.

**Strengths:**

•	Conceptual clarity: The drift penalty is cleanly derived from the linearization, tying representation drift to the GGN and enabling reuse of mature curvature approximations (Sec. 3.1–3.3).

•	Scalability: The merged KFAC surrogate (Eq. 8) achieves O(1) cost in tasks and empirically matches the naïve O(T) sum. Table 3 shows near-parity on ViT-B/16 and T5.

•	Strong empirical results: On 8-Vision, KFAC-regularized TA outperforms linear and non-linear FT, matches or beats τ-JP in several settings without external task data, and yields robust α=1 behavior (Table 1). It also improves task negation (Table 2).

•	Task-localization signal: The paper links its regularizer to a practical normalcy score (|J_\theta f(x,\theta_0)\tau_t|_2^2) that separates in- vs out-of-task inputs (Fig. 5).

•	Practicality: Overhead analysis and ablations (Fig. 6–7) are useful; MC=1 with ~128–256 examples often suffices for KFAC estimation.

**Weaknesses:**

•	Missing baseline: No comparison to Task-Localized Sparse FT; given the shared goal (localized updates with low interference), this absence limits claims of state-of-the-art effectiveness on task-local editing.

•	Attention-only baseline not fully aligned: While “Non-linear (Attn.)” appears, the paper doesn’t replicate the full protocol and metrics of Fine-Tuning Attention Modules Only, so the reader can’t conclude whether KFAC-regularized FT beats that specific method under its strengths (e.g., disentanglement maps/metrics across α-sweeps, stability under merging).

•	Heuristic merge lacks theory: The Kronecker aggregation (Eq. 8) is a heuristic; no approximation error bounds or conditions are provided, though it works empirically.

•	Memory footprint: The Limitations section concedes KFAC’s quadratic-in-units per layer memory cost, which could bite for very large LMs; compression or structured factors are left to future work.

•	Language results trail τ-JP: On T5, the approach narrows the gap but τ-JP still tops the leaderboard in places (Fig. 3a), hinting the curvature surrogate could be further improved for text.

•	Privacy/deployment details: Since the method proposes sharing curvature stats instead of data, the paper would benefit from concrete guidance on what to share (sizes, precisions, any leakage risks), especially beyond CLIP/T5 scales. (Code release is stated but details aren’t in the main text.)

**Questions:**

1.	TLSF baseline: Can you add results against Efficient Model Editing with Task-Localized Sparse Fine-tuning on the same 8-Vision and T5 setups, plus an editing-localization metric (e.g., Δ outside-task vs in-task) to demonstrate task localization superiority?

2.	Attention-only FT: Could you replicate the Fine-Tuning Attention Modules Only protocol (datasets, α-grids, disentanglement error maps) to enable a direct SOTA comparison under its claims?

3.	Theory for Eq. (8): Any conditions (e.g., approximate independence of factors across tasks) under which the Kronecker factor summation is a principled approximation to (\sum_t B_t\otimes A_t)?

4.	Memory/precision trade-offs: How do FP16/8-bit factor storage and block-sparsity affect accuracy and the normalcy score? Could you move factors off-GPU during training without bottlenecks?

5.	Scale to LLMs: What are the empirical costs when applying the method to 7B–70B-parameter LMs (rough factor sizes, setup time), and do your MC/num-examples findings (Fig. 7) still hold?

---

> ### Author Response · Authors · 2025-11-21
>
> We thank Reviewer 1ihd for the constructive review, as well as for the many concrete suggestions. We are grateful for the opportunity to clarify these points, many of which have been incorporated into the updated version of the manuscript.
>
> ## Missing baseline
>
> > Missing baseline: No comparison to Task-Localized Sparse FT ... this absence limits claims of state-of-the-art effectiveness on task-local editing.
>
> We have now added TaLoS (Task-Localized Sparse FT) as a baseline in the main tables. When comparing accuracy, our method consistently achieves **higher performance**. For instance, in terms of absolute accuracy:
>
> - ViT-B/32: 83.1 (ours, nonlinear regime) vs. 79.67 (TaLoS, +3.43)
> - ViT-B/16: 84.3 (ours, nonlinear regime) vs. 82.60 (TaLoS, +1.70)
> - ViT-L/14: 89.9 (ours, nonlinear regime) vs. 88.37 (TaLoS, +1.53)
>
> If we also consider the performance of our method in the linearized regime, the gap becomes even larger in favor of our approach. This behavior is further dissected in the updated Appendix (see Fig. 12 or the response to the next question), where we show that our regularizer yields models whose parameter updates are significantly more localized than those of TaLoS.
>
> > TLSF baseline: Can you add results against Efficient Model Editing with Task-Localized Sparse Fine-tuning on the same 8-Vision and T5 setups, plus an editing-localization metric (e.g., $\Delta$ outside-task vs in-task) to demonstrate task localization superiority?
>
> Following the reviewer’s recommendation, we conducted the editing-localization metric experiment ($\Delta$ outside-task vs. in-task) in the nonlinear regime for our method, TaLoS, and attention-only fine-tuning. The plots are reported in Fig. 12 of the updated manuscript. They clearly show that our method provides the best separation between in-distribution and out-of-distribution data, while TaLoS and attention-only approaches provide moderate but still meaningful improvements over the baseline.
>
> ## Comparison with Attention-only FT
>
> > While “Non-linear (Attn.)” appears, the paper doesn’t replicate the full protocol and metrics of Fine-Tuning Attention Modules Only, so the reader can’t conclude whether KFAC-regularized FT beats that specific method under its strengths (e.g., disentanglement maps/metrics across α-sweeps, stability under merging).
>
> We carefully re-examined the original Attention-Only Fine-Tuning paper to verify whether any methodological or evaluation differences existed. We did not identify discrepancies: our implementation follows the same protocol, and the resulting performance is essentially identical to the numbers reported in the original work. This strongly suggests that our reproduction is faithful and that the comparison was already sound.
>
> > Attention-only FT: Could you replicate the Fine-Tuning Attention Modules Only protocol (datasets, α-grids, disentanglement error maps) to enable a direct SOTA comparison under its claims?
>
> Following the suggestion, **we have now added both disentanglement-error maps** (Fig. 9) **and an $\alpha$-sweep experiment** (Fig. 11) in the updated Appendix. Regarding the disentanglement maps, although attention-only fine-tuning does improve over the full non-linear FT baseline (yielding a flatter landscape), our method remains clearly superior. A similar pattern emerges from the $\alpha$-sweeps (Fig. 11): our KFAC-regularized approach exhibits substantially higher stability across $\alpha$ values, indicating stronger robustness and better task-localized behavior, as also highlighted in Fig. 12 of the Appendix.
>
> ## Heuristic merge lacks theory
>
> > The Kronecker aggregation (Eq. 8) is a heuristic; no approximation error bounds or conditions are provided, though it works empirically.
>
> Also, in the **Questions** section:
>
> > Any conditions (e.g., approximate independence of factors across tasks) under which the Kronecker factor summation is a principled approximation to ($\sum_t B_t\otimes A_t$)?
>
> **We add a dedicated subsection in the Appendix** of the updated manuscript (Sec. C) where **we derive an approximation error bound** for the Kronecker merge. Specifically, we show that the error is bounded by $T \sigma_A \sigma_B$ where $\sigma_A$ and $\sigma_B$ denote the task-wise standard deviations of the $T$ K-FAC factors. Therefore, the approximation is tight whenever the K-FAC factors vary minimally across tasks.
>
> Importantly, we conjecture this matches the regime we operate in. Indeed, since all task-specific Fisher factors are estimated from the *same fixed pre-trained backbone* (e.g., CLIP), they all share a common underlying feature-space representation. As a consequence, both activation and gradient statistics $-$ and therefore the corresponding K-FAC factors $-$ exhibit limited task-to-task variability. According to our analysis, this is the underlying reason why the approximation remains close to the non-aggregated version, as already observed empirically in the original submission (Table 3, now Table 4).

---

> > ### Author Response · Authors · 2025-11-21
> >
> > ## Memory footprint
> >
> > > The Limitations section concedes KFAC’s quadratic-in-units per layer memory cost, which could bite for very large LMs; compression or structured factors are left to future work.
> >
> > Also, in the **Questions** section:
> >
> > > Memory/precision trade-offs: How do FP16/8-bit factor storage and block-sparsity affect accuracy and the normalcy score? Could you move factors off-GPU during training without bottlenecks?
> >
> > In the revised version of the main paper, **we added a dedicated experiment** that directly investigates these questions (see Fig. 8a and 8b).
> >
> > In a first set of experiments, performed on ViT-B/16 in the 8-Vision setup, we applied several compression techniques to the KFAC matrices and measured their impact on avg. abs. accuracy. Specifically, we evaluated: dynamic 8-bit quantization, pruning with keep rates of 0.30 and 0.15, a block-diagonal strategy with 8 blocks, and truncated SVD. Each method is described in detail in the supplementary materials.
> >
> > The main finding is that **memory usage can be substantially reduced with limited performance degradation**, as shown in Fig. 8a of the updated manuscript. The best trade-off is achieved by the block-based approximation, which reduces memory **from roughly 550 MB (full KFAC) to about 70 MB**. Namely, we can achieve a memory reduction of approximately 87% $-$ with only a $\approx$ **1-point drop in accuracy** (from 88.40 to 87.12).  For the block-based strategy, we also reproduced the task-localization plots, reported in Fig. 13 of the updated supplementary materials. This analysis shows that using only 8 diagonal blocks has negligible effect on this metric: the model still cleanly separates in-distribution and out-of-distribution samples.
> >
> > > Could you move factors off-GPU during training without bottlenecks?
> >
> > Yes, our method is compatible with a scheme in which the KFAC factors are dynamically moved on and off the GPU as needed. Naturally, if the regularization loss must be computed at every iteration, this introduces overhead. For this reason, in a second experiment (Fig. 8b of the updated manuscript), we tested a regime where the regularization loss is computed and backpropagated **only once every *N* training steps**.
> >
> > As shown in the results, even when the loss is applied only every 16 steps, the performance degradation is limited ($\approx$ 1.4 points worse than applying the loss at every step) while still yielding a gain of about 5 points over the baseline that never applies the loss. This suggests that infrequent regularization updates can effectively amortize memory transfers without significantly compromising accuracy.
> >
> > ## Language results trail $\tau$JP
> >
> > > On T5, the approach narrows the gap but τ-JP still tops the leaderboard in places (Fig. 3a), hinting the curvature surrogate could be further improved for text.
> >
> > We agree that, on T5, there is still room to improve the quality of the curvature surrogate, and we believe that such improvements would likely translate into further gains in task arithmetic performance. In fact, one of the main contributions of our work is precisely to bridge curvature estimation and task-vector quality, enabling future advances in curvature modeling to directly benefit task arithmetic. We see our method as laying the groundwork for this connection, and we expect more refined curvature estimators to push performance even further.
> >
> > ## Privacy/Deployment Details/Application to LLMs
> >
> > > Privacy/deployment details: Since the method proposes sharing curvature stats instead of data, the paper would benefit from concrete guidance on what to share (sizes, precisions, any leakage risks), especially beyond CLIP/T5 scales. (Code release is stated but details aren’t in the main text.)
> >
> > **Privacy.** We share one *aggregated* KFAC package per backbone: *i.e.*, per-layer pairs $(A_\ell,B_\ell)$ as float tensors. These are the covariances of layer inputs and output gradients, which avoid the exposure to training data. This is also confirmed in federated learning settings, where activations and gradients are often exchanged even under critical privacy requirements [3, 4]. Some studies explicitly attempt to reconstruct the original inputs from the shared activations, showing that this is not possible [5].
> >
> > **Size of KFAC (fp32).** The KFAC factors that we use within our work have the following memory footprint:
> > - ViT-B (patch 16 or 32) $\approx$ 550 MB;
> > - ViT-L/14 $\approx$ 2.7 GB;
> > - T5-base $\approx$ 1.2GB.
> >
> > Larger language models such as Llama-3B would require $\approx$ 8 GB. However, within attention-only fine-tuning $-$ which shows comparable results to the linearized regime $-$ the memory footprint reduces drastically, bringing the required storage for Llama-3B to $\approx$ 4.5 GB.

---

> > > ### Author Response · Authors · 2025-11-21
> > >
> > > > Scale to LLMs: What are the empirical costs when applying the method to 7B–70B-parameter LMs (rough factor sizes, setup time), and do your MC/num-examples findings (Fig. 7) still hold
> > >
> > > Since KFAC scales quadratically with each layer’s activation widths (input and output), the required fp32 storage becomes substantial: for example, $\approx$ 12 GB for LLaMA-7B and $\approx$ 120 GB for LLaMA-70B. For models of this scale, more efficient alternatives could be explored, such as operating in the pseudo-linear regime and fine-tuning only the attention layers. Moreover, the 8-block compression strategy introduced in the updated supplementary materials further reduces the memory footprint of the KFAC factors by a large margin.
> > >
> > > Finally, we were unable to run experiments on models above 3B parameters within the time constraints of the rebuttal, so we cannot empirically verify whether the trends observed in Fig. 7 persist at larger scales. However, the results are consistent across all tested vision and language models, suggesting that the underlying relationship between number of samples and performance is stable within the explored range. This is also supported by the KFAC literature on single-sample curvature estimation [1, 2].
> > >
> > > [1] Martens, J., & Grosse, R. (2015, June). Optimizing neural networks with kronecker-factored approximate curvature. In ICML.
> > >
> > > [2] Eschenhagen, R., et al. (2023). Kronecker-factored approximate curvature for modern neural network architectures. In NeurIPS.
> > >
> > > [3] Tan, Y., et al. (2022). Fedproto: Federated prototype learning across heterogeneous clients. In AAAI.
> > >
> > > [4] Luo, K., et al. (2023). Gradma: A gradient-memory-based accelerated federated learning with alleviated catastrophic forgetting. In CVPR.
> > >
> > > [5] Luo, M. et al., (2021). No fear of heterogeneity: Classifier calibration for federated learning with non-iid data. In NeurIPS.

---

> > > > ### Comment · Reviewer_1ihd · 2025-11-26
> > > > **Records of changes**
> > > >
> > > > Dear authors, thank you for your detailed responses. Could you please highlight your changes in the manuscript so that I can easily figure them out? Many thanks.

---

> > > > > ### Author Response · Authors · 2025-11-26
> > > > >
> > > > > Thank you for your response. We have now uploaded an updated version of the manuscript where **all changes are highlighted in blue**.
> > > > >
> > > > > If helpful, we also provide below a brief summary of the main changes introduced in the manuscript to address your concerns.
> > > > >
> > > > > ---
> > > > >
> > > > > ### Summary of Changes
> > > > >
> > > > > - **Added missing baseline (TaLoS).** TaLoS has been added to all main tables. Our method consistently outperforms it across ViT-B/32, ViT-B/16, and ViT-L/14. A detailed analysis of editing localization (Δ outside-task vs. in-task) is now provided in Fig. 12.
> > > > >
> > > > > - **Strengthened comparison with Attention-Only Fine-Tuning.** We verified full protocol alignment and added the requested experiments: disentanglement-error maps (Fig. 9) and $\alpha$-sweeps (Fig. 11). Our method shows higher stability and better task localization.
> > > > >
> > > > > - **Added theoretical justification for the Kronecker merge.** A new Appendix section (Sec. C) derives an approximation error bound, clarifying when the aggregation is principled ($\rightarrow$ low task-wise variability of factors).
> > > > >
> > > > > - **Expanded memory analysis and compression strategies.** We added new experiments (Fig. 8a–b) evaluating 8-bit quantization, pruning, block-diagonal factors, and truncated SVD. The block-based method reduces memory from **550 MB to ~70 MB** with only ≈1-point accuracy loss, and maintains task localization (Fig. 13). We also evaluate computing the regularization loss only every *N* steps to reduce GPU transfers.
> > > > >
> > > > > - **Remaining comments** (language results, privacy/deployment, and LLM scaling) are also addressed in detail in the response above.
> > > > >
> > > > > ---
> > > > >
> > > > > Please let us know if any further clarification would be helpful.

---

> > > > > > ### Comment · Reviewer_1ihd · 2025-11-27
> > > > > > **Update**
> > > > > >
> > > > > > The authors' responses addressed most of my concerns, I'd like to recommend for acceptance.

---

> > > > > > > ### Author Response · Authors · 2025-11-27
> > > > > > >
> > > > > > > Thank you very much for your positive assessment and for recommending the paper for acceptance. We truly appreciate the time and care you dedicated to the review process.
> > > > > > >
> > > > > > > We remain fully available to provide any further clarification or additional material, both to you and to the other reviewers.

---

### Official Review · Reviewer_KRUA · 2025-11-01

**Soundness:** 2
**Presentation:** 3
**Contribution:** 1
**Rating:** 2
**Confidence:** 4

**Summary:**

The paper addresses cross-task interference in model merging/task arithmetic by proposing a method for enhancing weight disentanglement. The authors demonstrate that *representation drift regularization*, a data-dependent solution, can be reformulated in a linearized setting as a quadratic form involving the Jacobian's Gramian matrix (G). They then use a tractable approximation (KFAC) of G to propose a new, data-free regularizer. The authors also propose a heuristic to merge KFAC matrices with O(1) complexity. The resulting KFAC-regularized task vectors are shown to be more disentangled, leading to better performance when merging models that have been fine-tuned linearly.

**Strengths:**

- The theoretical derivation is elegant, connecting representation drift in a linearized model to a practical, data-free regularizer via the GGN and KFAC.
- In the linearized fine-tuning setting, the proposed method performs on-par with or better than the data-dependent $\tau$Jp baseline.
- KFAC regularization at training time allows simple weight averaging (task arithmetic) to outperform more complex, SOTA post-hoc merging methods (like TSV) that are applied to unregularized vectors if training was in the linearized regime.
- The method is shown to be computationally fast.

**Weaknesses:**

- The entire experimental validation is confined to the linearized finetuning framework (see, e.g., Ortiz-Jimenez et al., 2023). The authors don't test their KFAC regularizer on full, non-linear finetuning. Is the regularizer only effective in a linearized regime?
- The paper's validation against SOTA merging methods (TIES, TSV, etc.) is limited to this niche linearized FT setting (Figure 4). This avoids the most practical question: how do these methods perform on task vectors from standard, non-linear fine-tuning? These non-linear task vectors are the default, readily available artifacts in most real-world scenarios. If SOTA post-hoc methods applied to these standard vectors already outperform the entire KFAC/linearized-FT framework, the practical motivation for this work is unclear. The paper fails to demonstrate that its regularizer is necessary or effective in the standard, non-linear setting.
- (Minor) The performance gains on T5 are less impressive, and the data-dependent alternative ($\tau$Jp) is better.

**Questions:**

- What are the performance results of SOTA post-hoc methods (TIES, TSV, etc.) when applied to task vectors from standard non-linear fine-tuning? How does the KFAC-regularized framework compare in that more practical scenario?

---

> ### Author Response · Authors · 2025-11-21
>
> We thank Reviewer KRUA for their thoughtful assessment, and for highlighting both the strengths of our theoretical derivation. We also apologize if some aspects of our method’s application to the non-linear fine-tuning regime were not communicated clearly enough in the original submission.
>
> ## Applications to the non-linear FT regime
> > **“The entire experimental validation is confined to the linearized finetuning framework (see, e.g., Ortiz-Jimenez et al., 2023). The authors don't test their KFAC regularizer on full, non-linear finetuning. Is the regularizer only effective in a linearized regime?”**
>
> We clarify that **our validation is not restricted to the linearized fine-tuning setting**. The originally submitted manuscript already includes experiments where our KFAC regularizer is integrated into a standard non-linear fine-tuning pipeline, and these results show **clear and tangible performance** gains.
>
> For example, Fig. 2 and Fig. 3 of the original manuscript (right radar plots) report results obtained under non-linear FT, with our method consistently outperforming all competitors. Table 4 (original submission) provides the corresponding numerical values, showing substantial absolute-accuracy gains over Attention-Only Fine-Tuning (Jin et al., 2025): on ViT-B/32 we obtain 83.1 vs. 78.2 (+4.9), on ViT-B/16 we obtain 84.3 vs. 80.4 (+3.9), and on ViT-L/14 we obtain 89.9 vs. 88.2 (+1.7).
>
> In these settings, we combine our KFAC regularizer with Attention-Only Fine-Tuning, a scheme shown by Jin et al. (2025) to induce approximately linear behavior even within a nonlinear fine-tuning regime. In our approach, this provides a practical way to ensure that the linearized approximation remains accurate while still operating in the standard nonlinear setting. In turn, this validates both the mathematical grounding and the empirical effectiveness of our method beyond the strictly linearized regime.
>
> In the revised version, we explicitly label the non-linear results and move them from the supplementary into the main paper for clarity. To further strengthen this point, we additionally evaluated task localization (Fig. 5) in the non-linear regime. As shown in Fig. 12 of the updated supplementary materials, our method continues to promote well-separated, task-specific activation patterns even under non-linear FT. This demonstrates that our KFAC-based regularization effectively encourages localized updates aligned with the training distribution, a key property underlying weight disentanglement.
>
> ## Comparison with non-linear approaches
>
> >The paper's validation against SOTA merging methods (TIES, TSV, etc.) is limited to this niche linearized FT setting (Figure 4). This avoids the most practical question: how do these methods perform on task vectors from standard, non-linear fine-tuning? These non-linear task vectors are the default, readily available artifacts in most real-world scenarios. If SOTA post-hoc methods applied to these standard vectors already outperform the entire KFAC/linearized-FT framework, the practical motivation for this work is unclear.
>
> Before addressing the reviewer’s question, we emphasize that there are at least three practical motivatios why our approach remains advantageous compared to applying post-hoc merging methods (TIES, TSV, etc.). In brief, our curvature-regularized training produces task vectors that are explicitly **weight disentangled**, substantially **more robust** to the choice of $\alpha$, and supported by a formulation that remains **analytically tractable** $-$ something that is considerably harder to achieve in the fully nonlinear regime. Last but not least, as we show a few lines below, our method performs on par with state-of-the-art approaches such as TSV.
>
> **Weight disentanglement.** It is important to clarify that our work addresses *Task Arithmetic*, not model merging, thereby supporting a richer family of operations on task vectors beyond merging. For instance, our method enables meaningful **subtraction** of task vectors, i.e., enabling zero-shot unlearning of pretraining concepts. Post-hoc model-merging methods such as TSV and TIES do not produce disentangled task vectors, cannot support effective task negation, and their behavior is fundamentally constrained by the entanglement introduced by nonlinear fine-tuning.
>
> **Robustness to the choice of $\alpha$**. We showed at multiple points in the paper that our approach is substantially more robust to the choice of $\alpha$ $-$ i.e., the hyperparameter that scales the merged task vector (see first paragraph of Sec. 4). This robustness is important in practice because it eliminates the need to access validation data from other tasks, whereas post-hoc nonlinear merging methods such as TSV explicitly require the validation set of *each* task to tune $\alpha$ effectively.

---

> > ### Author Response · Authors · 2025-11-21
> >
> > Following the reviewer’s suggestion, we performed a direct comparison with the state-of-the-art TSV-M method, explicitly evaluating its robustness to $\alpha$ using task vectors obtained from standard non-linear fine-tuning. To make the comparison as fair as possible, we evaluated TSV-M both by reimplementing it within our own codebase and by using the authors’ official repository and released checkpoints. The two TSV variants yielded slightly different results, and we believe this discrepancy is due to environment incompatibilities (our framework cannot run on the older PyTorch version used in their implementation).
> >
> > **Result: TSV needs $\alpha$-tuning on the validation set to work well, while our method stays strong across all $\alpha$ without any tuning.**
> >
> > ### **Alpha Sweep TSV ViT-B/16 absolute accuracy**
> > | METHOD \ $\alpha$ | 0.1 | 0.2 | 0.3 | 0.4 | 0.5 | 0.6 | 0.7 | 0.8 | 0.9 | 1.0 | 1.1 | 1.2 | 1.3 | 1.4 | 1.5 |
> > | --- | --- | --- | --- | --- | --- | --- | --- | --- | --- | --- | --- | --- | --- | --- | --- |
> > | TSV - Original implementation | 61.4| 68.4|74.0|78.4 |81.8|84.5|86.5|87.8| 88.4|88.8|88.6|88.4|87.8|87.1|86.0|
> > | TSV - Our implementation | 62.7|69.5 | 74.5| 78.8| 81.9| 84.0|85.4 |86.3 | 86.8|87.1|87.0| 86.7|86.3| 85.5| 84.3|
> > | Ours - KFAC |79.7|84.9 | 86.8|87.7| 88.0|88.3 | 88.4|88.4 |88.4 |88.4| 88.4| 88.4|88.4 | 88.4| 88.4|
> >
> > ### **Alpha Sweep TSV ViT-B/16 normalized accuracy**
> > | METHOD \ $\alpha$ | 0.1 | 0.2 | 0.3 | 0.4 | 0.5 | 0.6 | 0.7 | 0.8 | 0.9 | 1.0 | 1.1 | 1.2 | 1.3 | 1.4 | 1.5 |
> > | --- | --- | --- | --- | --- | --- | --- | --- | --- | --- | --- | --- | --- | --- | --- | --- |
> > | TSV - Original implementation |67.1 | 74.4|80.2| 84.9|88.6|91.5|93.3|95.1| 95.7|96.2|96.0|95.7|95.0|94.1|92.7|
> > | TSV - Our implementation | 68.3| 75.3| 80.5|85.1 |88.4|90.7 |92.2|93.2|93.7|94.0|93.9|93.5|92.9|92.0|90.7|
> > | Ours - KFAC | 88.0|93.8|96.0 | 97.1|97.5|97.8 |97.9|97.9| 98.0|98.0|97.9 | 97.9| 97.9| 97.9| 97.9|
> >
> > While both methods reach similar peak accuracy when $\alpha$ is finely tuned, our approach sustains high performance over a much broader $\alpha$ range. In practice, this robustness is crucial $-$ TSV is highly sensitive to $\alpha$ and requires a separate validation set for each task to tune it reliably.
> >
> > **Analytical tractability.** Additionally, the linearized regime allows for mathematically tractable analysis and provides a clearer foundation for studying weight disentanglement, whereas standard full fine-tuning makes such analysis significantly more complex.
> >
> > >**“The paper's [...] is limited to this niche linearized FT setting”**
> >
> > We respectfully clarify that the linearized regime is far from a marginal or isolated setting. It has recently developed into a substantial research direction spanning NTK theory, tangent-space task arithmetic, and kernel interpretations of Transformers. To contextualize this line of work, we added a short overview in the supplementary material (App. G). Collectively, these contributions total well over 7,000 citations and include more than 15 papers across top-tier venues such as NeurIPS, ICLR, ICML, CVPR, ECCV, and ICCV.
> >
> > ## Performance
> >
> > > (Minor) The performance gains on T5 are less impressive, and the data-dependent alternative (Jp) is better.
> >
> > This finding is unsurprising, since our method provides a data-free approximation of the same regularization objective optimized by τJp, which instead requires raw data from the other tasks (whereas we rely on KFAC statistics). Despite this difference, our approach remains very close in performance (e.g., 98.9 vs. 100 norm. accuracy, Figure 3).
> >
> > Moreover, the small accuracy gap observed on textual tasks should be interpreted in light of the substantially **superior computational efficiency** of our method. As shown in Figure 6a, a single forward–backward pass with our approach takes roughly **1.3 seconds**, compared to **3.2 seconds** for τJp using the same GPU.
> >
> > In addition, our method is inherently more data-flexible: it does not require any transfer of samples across compute nodes, nor coordination of per-task batches during training. This makes it considerably easier to deploy in distributed environments.

---

### Author Response · Authors · 2025-12-02
**Summary of the rebuttal process**

We thank the Area Chair for the tremendous work they have done under such challenging circumstances. Below, we provide a brief summary of the review process, focusing on the major points raised by the reviewers, although in the rebuttal we addressed all of their concerns and questions.

## Reviewer KRUA
*(originally recommended rejection; **no responses** during the rebuttal period)*

*“The entire experimental validation is confined to the linearized fine-tuning framework.”*
We pointed to several sections of the original submission showing that our approach is successfully applied in the non-linear fine-tuning regime.

*“If SOTA post-hoc methods already outperform the entire KFAC/linearized-FT framework, the practical motivation for this work is unclear.”*
We highlighted that our method greatly enhances weight disentanglement (e.g., unlearning) and robustness to the hyperparameter $\alpha$. A similar concern was raised by reviewer K5Dr, who ultimately responded positively, stating that all concerns were resolved.

*“The performance gains on T5 are less impressive, and τJp performs better.”*
We note that τJp relies on raw data from other tasks, whereas our method is data-free; despite this, we achieve comparable performance and substantially faster training steps (1.3s vs. 3.2s for τJp).

## Reviewer 1ihd
*(after the rebuttal period recommended acceptance — **“The authors' responses addressed most of my concerns, I'd like to recommend for acceptance.”**)*

*“Missing baseline.”*
We added TaloS in both the main tables and the experimental analysis, showing that our approach consistently outperforms it, and we included the requested comparisons with Attention-only FT.

*“Heuristic merge lacks theory.”*
We added a new Appendix section (Sec. C) deriving an approximation error bound.

*“Lacking experiments with compressed KFAC.”*
We evaluated our approach with several compression techniques, showing that memory can be reduced from 550 MB to ~70 MB with only $\approx$1-point accuracy loss.

## Reviewer RdRs
*(originally voted borderline accept; **no responses** during the rebuttal period)*

*“Novelty (minor).”*
We clarified the novel aspects of our work, which were later explicitly appreciated by reviewer K5Dr.

*“Questions.”*
We addressed all the questions raised (notation, relations to prior work, typos, etc.).

## Reviewer K5Dr
*(after the rebuttal period recommended acceptance — **“All of my concerns are now resolved, and I feel more positive about acceptance... Thank you for the excellent work and the careful rebuttal.”**)*

*“The approach requires KFAC from other tasks.”*
We showed that KFAC statistics computed on ImageNet or related publicly available datasets can be used with only a small performance drop.

*“Missing non-linear regime comparisons to SOTA.”*
We added comparisons with SOTA post-hoc methods, showing that our approach is competitive, more robust under $\alpha$-sweep analyses, and produces more disentangled task vectors (useful for task arithmetic).

*“Lacking sensitivity analysis.”*
We provided this analysis, demonstrating that our method remains stable across a wide range of hyperparameter values.

---

### Meta-Review · Area_Chair_n88s · 2026-01-13

**Summary:**

1. The authors don't test their KFAC regularizer on full, non-linear finetuning. The paper's validation against SOTA merging methods (TIES, TSV, etc.) is limited to this niche linearized FT setting.
2. No comparison to Task-Localized Sparse FT.
3. The Limitations section concedes KFAC’s quadratic-in-units per layer memory cost, which could bite for very large LMs.
4. On T5, the approach narrows the gap but τ-JP still tops the leaderboard in places (Fig. 3a), hinting the curvature surrogate could be further improved for text.

**Reviewer Concerns:**

Addressed concerns:
1. The authors clarify that our validation is not restricted to the linearized fine-tuning setting. The originally submitted manuscript already includes experiments where our KFAC regularizer is integrated into a standard non-linear fine-tuning pipeline, and these results show clear and tangible performance gains. The authors also add a direct comparison with the state-of-the-art TSV-M method.
2. The authors added TaLoS (Task-Localized Sparse FT) as a baseline in the main tables.
3. The authors added a dedicated experiment that memory usage can be substantially reduced with limited performance degradation, but with 1-point drop in accuracy.

Outstanding concerns:

4. The authors admit that on T5, there is still room to improve the quality of the curvature surrogate.

**Reviewer Scores:**

Reviewer 1ihd recommend for acceptance and may raise the score to 6-7.

---

### Decision · Program_Chairs · 2026-01-26

Accept (Poster)